# A multilayered genetic structure analysis between inflammatory bowel disease and bone density/osteoporosis

Mengting Qin[1]*, Xinhong Liu[2], Qinghua Luo[3], Ruyun Cai[1]

1 Department of Anorectal Surgery, Zhejiang Chinese Medical University Affiliated Jiaxing TCM Hospital, Jiaxing, China, 2 Department of Anorectal Surgery, Affiliated Hospital of Jiangxi University of Chinese Medicine, Nanchang, China, 3 Clinical Medical College, Jiangxi University of Chinese Medicine, Nanchang, China

* qinmt0907@163.com

## Abstract

### Background

The association between inflammatory bowel disease (IBD) and reduced bone mineral density (BMD) or osteoporosis is still a subject of ongoing debate, underscoring the need for further exploration, particularly from a genetic perspective.

### Methods

Utilizing data from genome-wide association studies on 59,957 IBD, 40,266 CD, 45,975 UC, 31,492 BMD, and 399,054 osteoporosis, comprehensive analyses were performed focusing on two main aspects: genetic correlations (Rg) and shared genetic loci. Initially, the overall Rg between these traits was assessed via genetic covariance analyzers and high-definition likelihood approaches. Following this, local genetic patterns were examined using local variant association analysis. Mendelian randomization (MR) analysis was performed to infer potential causal relationships. The genetic overlap was then explored using conditional/conjunctional false discovery rate (cond/conjFDR) statistical methods, leading to the identification of several biologically significant shared genetic loci.

### Results

A notable genetic correlation was found between IBD, its subtypes, and BMD/osteoporosis at the genome-wide level. Additionally, significant local genetic associations were identified across various chromosomal regions. The conjFDR analysis further supported the genetic overlap and pinpointed several critical shared loci. Furthermore, significant enrichment of the Wnt signaling pathway was detected for both conditions.

**Data availability statement:** All relevant data are within the paper and its Supporting Information files.

**Funding:** This study was supported by the Jiangxi Provincial Natural Science Foundation project (20232BAB206154) and the University-Level Research Project of Zhejiang Chinese Medical University (2022FSYYZQ21). The funders had no role in study design, data collection and analysis, decision to publish, or preparation of the manuscript.

**Competing interests:** The authors declare that the research was conducted in the absence of any commercial or financial relationships that could be construe as a potential conflict of interest.

## Conclusion

This investigation offers robust genetic evidence supporting the comorbidity of IBD with BMD/osteoporosis, highlights possible underlying mechanisms, and provides new insights into clinical research and practice.

## 1. Introduction

Inflammatory bowel disease (IBD) is a chronic gastrointestinal disorder primarily triggered by abnormalities in the immune system, typically classified into Crohn's disease (CD) and ulcerative colitis (UC) [1]. Common manifestations include abdominal pain, diarrhea, bloody stools, and weight loss [2]. Research has highlighted the significant roles played by the intestinal environment, microbiota dysbiosis, and genetic factors in its pathogenesis, although the exact mechanisms remain incompletely understood [3]. With the rapid pace of global industrialization, the incidence of IBD has been steadily rising. Recent 2025 studies show that IBD has entered a "high prevalence stage," with both incidence and prevalence at elevated levels. In Western countries, the prevalence has reached or is near 1%, and by 2030, over 1% of the population is expected to be affected, putting significant strain on healthcare systems [4]. IBD patients may endure lifelong challenges from ongoing inflammation and related complications, one of which is reduced bone mineral density (BMD), thereby heightening the risk of osteoporosis [5–7]. Recent Mendelian randomization studies have revealed significant causal relationships between IBD and both osteoporosis and BMD [8–10]. Bone is a dynamically remodeling tissue maintained through the balance between osteoblasts and osteoclasts [11]. Osteoporosis occurs when bone resorption outpaces bone formation, leading to a decline in bone mass and structural integrity, which in turn increases fracture risk [12].

Osteoporosis affected 41.5 million people globally in 2019, with a projected 263.2 million cases by 2034 [13]. With an aging population, the prevalence of osteoporosis continues to grow, earning the distinction of being termed the "silent disease of the 21st century". The potential shared genetic basis between IBD, BMD, and osteoporosis provides a critical avenue for future research, particularly from the perspective of comorbidity genetics.

In recent years, the advancement of genome-wide association studies (GWAS) has laid a solid foundation for the investigation of genetic associations among IBD, BMD, and osteoporosis. Concurrently, several novel and reliable genetic statistical methods have emerged, enabling more comprehensive research in this domain.

The following analytical tools are specifically designed to estimate the degree of overall and localized shared genetic correlation (Rg) between complex traits. The genetic covariance analyzer (GNOVA) is one such tool; it estimates genetic covariance based on GWAS data, providing efficient, robust annotated stratification analysis capabilities, which helps uncover the genetic architecture of complex traits [14]. High-definition likelihood (HDL) is a full likelihood-based approach to estimate genetic correlations (Rg); it utilizes linkage disequilibrium (LD) information derived from

GWAS data, offering higher precision and identifying a broader range of significant correlations compared to conventional methods [15]. The local analysis of variant association (LAVA) is used for local Rg analysis, assessing shared genetic bases between traits in specific genomic regions, supporting multivariate analysis and effectively handling sample overlap [16]. For example, HDL has confirmed significant genetic correlations between antisocial behavior and psychiatric disorders [17], and LAVA analysis has been extensively used to investigate shared genetic mechanisms between cardiovascular disease and schizophrenia [18].

The conditional false discovery rate (condFDR) method, and its derivatives, are used to identify shared genetic loci and quantify genetic overlap. The condFDR method improves the efficiency of identifying trait-associated genetic loci by incorporating cross-trait genetic enrichment, thereby reprioritizing GWAS statistics [19]. The conjunctional false discovery rate (conjFDR) is built upon this framework to precisely pinpoint shared genetic loci, thus unveiling common genetic features across complex traits [19]. Moreover, the genetic overlap between IBD and systemic lupus erythematosus has been robustly validated through the application of GNOVA and conjFDR methods [20].

This study seeks to bridge the existing knowledge gap concerning the genetic links between IBD and BMD/osteoporosis, with a focus on two principal objectives: assessing the Rgs between these conditions and pinpointing shared genetic loci. To accomplish these aims, genome-wide Rgs were examined using GNOVA [14] and HDL [15] methodologies, while local Rgs were explored through LAVA [16]. Mendelian randomization (MR) analysis was performed to infer potential causal relationships [21]. The identification of shared loci was facilitated by the use of the cond/conjFDR [17] method, which is extensively applied in genetic research involving comorbid traits. The findings of this study are anticipated to offer pivotal scientific insights into the genetic connections between IBD and BMD/osteoporosis, potentially opening new directions for genomic and precision medicine investigations.

## 2. Methods and materials

### 2.1. GWAS data

In the process of screening GWAS data related to IBD, BMD, and osteoporosis, the criteria for selection were primarily centered on sufficient sample size, comprehensive single nucleotide polymorphism (SNP) coverage, and the inclusion of recent data updates. The GWAS results for IBD ($N_{case} = 25,042$, $N_{control} = 34,915$, $N_{total} = 59,957$) and its subtypes (CD ($N_{case} = 12,194$, $N_{control} = 28,072$, $N_{total} = 40,266$) and UC ($N_{case} = 12,366$, $N_{control} = 33,609$, $N_{total} = 45,975$)) were sourced from the study by de Lange KM et al. [22]. Data pertaining to BMD was obtained from a meta-analysis examining total BMD ($N_{total} = 31,4921$) and age-related effects [23]. The osteoporosis data ($N_{case} = 8,017$, $N_{control} = 391,037$, $N_{total} = 399,054$) were extracted from the FinnGen database (https://r10.finngen.fi/) [24].

To ensure the reliability and validity of the genetic analysis results, we implemented a strict quality control process. Variants included in the final analysis had to meet the following criteria: First, variant screening and calibration were based on the 1000 Genomes Project Phase 3 data as the reference panel. Second, only biallelic variants were retained, with the minor allele frequency (MAF) for European population data set above 0.01 to ensure statistical power. Finally, all variants were accurately annotated to their genomic positions based on the human reference genome (hg19/GRCh37). Variants lacking rsID identifiers or with conflicting rsID annotations were excluded to ensure data accuracy.

To rigorously control the false-positive rate in the analysis, this study applied specific statistical correction strategies for each method. For the genome-wide genetic correlation analyses conducted using GNOVA, HDL, and MR, we applied the Bonferroni correction. In the case of LAVA analysis, p-values were adjusted using the Benjamini-Hochberg (BH) method to effectively control the false discovery rate (FDR) and identify significant associations. In the condFDR/conjFDR analyses, a significance threshold of conjFDR < 0.05 was adopted, consistent with the practices commonly used in previous studies [25–28].

All GWAS summary statistics used in this study originated from previously published datasets that had obtained institutional ethical approval, participant consent, and underwent strict quality control. All analyses were conducted using

populations of European ancestry. The detailed characteristics of the cohort are shown in <u>Table 1</u>. The flow chart of this study is shown in <u>Fig 1</u>.

## 2.2. Global Rg analyses

The overall Rg between two traits is calculated by GNOVA through the method of moments framework [14]. Initially, it is assumed that GWAS data for both traits share a common SNP list, reflecting the link between genotypes and phenotypes via a standardized linear model. Subsequently, GNOVA constructs a series of matrices that connect GWAS z-score matrices with genetic covariance parameters. These matrices are employed by GNOVA to develop a system of linear equations for calculating the covariance between SNP z-scores and genetic effects. The genetic covariance estimates between traits are derived by solving these equations, with adjustments made for sample overlap and non-genetic noise. In the final step, GNOVA incorporates external reference panels (such as the 1000 Genomes Project) to calculate LD, thereby optimizing the estimation outcomes and ensuring both computational accuracy and robustness.

The overall Rg between two traits is estimated by HDL through a full likelihood approach [15]. The procedure involves several steps: Initially, a phenotypic covariance matrix model is constructed by using the z-score matrix derived from

**Table1.  Date sources.**

| Phenotypes | Phenotypic code | Cases/Controls | Ancestry |
|---|---|---|---|
| IBD | ebi-a-GCST004131 | 25,042/34,915 | European |
| CD | ebi-a-GCST004132 | 12,194/28,072 | European |
| UC | ebi-a-GCST004133 | 12,366/33,609 | European |
| BMD | ebi-a-GCST005348 | NA/NA | European |
| osteoporosis | M13_OSTEOPOROSIS | 8,017/391,037 | European |

IBD: Inflammatory bowel disease; CD: Crohn's disease; UC: Ulcerative colitis; BMD, bone mineral density.

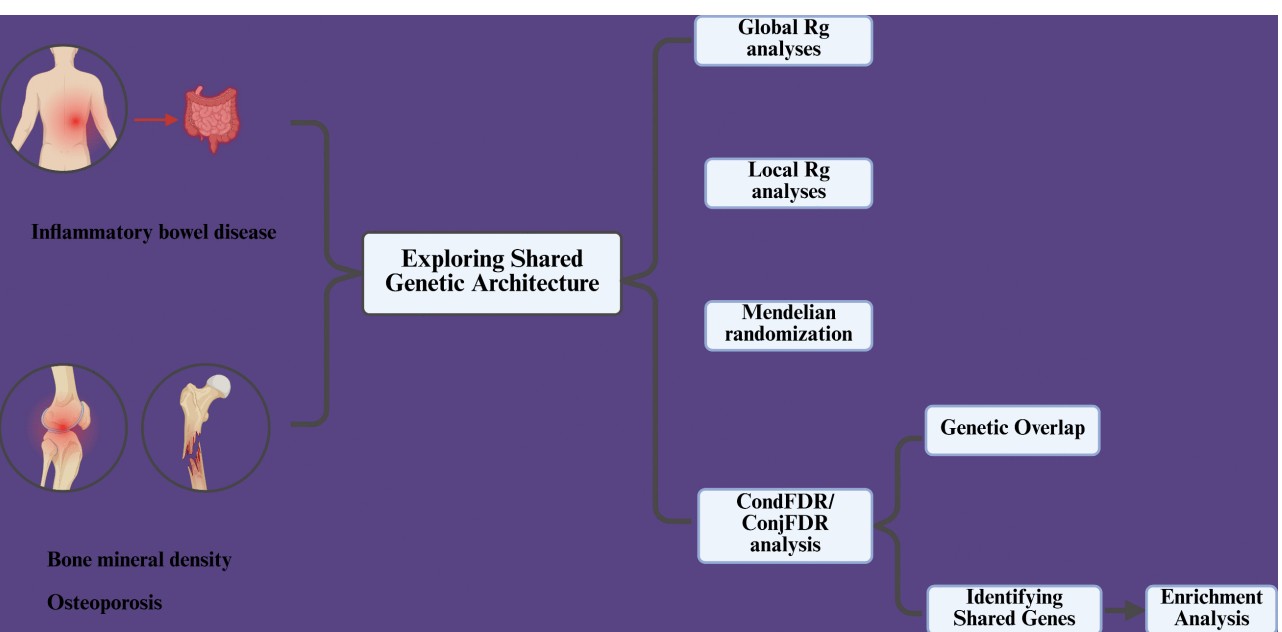

**Fig 1.  Flowchart of the study.** GNOVA,genetic covariance analyzers; HDL, high-definition likelihood; LAVA, local variant association.

GWAS data and the LD matrix, which is calculated using 307,519 SNPs from 336,000 genetically unrelated British individuals in the UK Biobank. This matrix links genetic covariance parameters with observed data. Next, HDL incorporates all available LD information (including off-diagonal elements) to concurrently infer genetic variance, genetic covariance, and correlation between traits using maximum likelihood estimation. In contrast to alternative methods, such as linkage disequilibrium score regression, HDL fully utilizes genome-wide LD information, thereby enhancing both estimation accuracy and statistical power, particularly for traits with low heritability.

The Rg functions as a critical measure of the extent to which shared genetic variants exist between two traits, with values spanning from −1–1. Specifically, an Rg value of 1 signifies a perfect positive correlation between the traits, whereas an Rg of 0 indicates the absence of a genetic relationship, and an Rg of −1 denotes a perfect negative correlation. By analyzing Rgs across various traits, the underlying genetic architecture can be comprehensively explored, thereby uncovering genetic links between diseases and potentially shared biological mechanisms.

### 2.3. Local Rg analyses

The local Rg between two traits is calculated by LAVA through the following procedure: First, within user-defined genomic regions (1Mb blocks), local genetic effects for each trait are estimated using GWAS summary statistics by constructing multivariate linear regression models that associate standardized phenotypes with combined SNP effects [16]. To identify chromosomal segments associated with IBD and BMD, LAVA selects specific genomic regions based on predefined 1Mb blocks. These blocks are defined by genomic coordinates, and their selection is optimized to minimize LD between adjacent blocks, ensuring independent genetic signals are captured. In LAVA analysis, the LD matrix used is based on the data from the European panel of the 1000 Genomes Project (Phase 3). Subsequently, local genetic covariance matrices are derived using the method of moments, from which covariance and genetic variance for each trait pair are extracted. Utilizing the covariance matrices, local Rgs are determined, and their significance is assessed. Computational accuracy is ensured by LAVA through the incorporation of LD information and reference data, while the robustness of the analysis is enhanced by correcting for sample overlap. Furthermore, LAVA facilitates multivariate analysis, allowing for the exploration of conditional genetic relationships among multiple traits, thereby yielding deeper insights into complex genetic mechanisms.

### 2.4. Mendelian-randomization analysis

This study utilized a bidirectional two-sample Mendelian randomization (MR) approach to explore the causal relationships between IBD, BMD, and osteoporosis. The analysis strictly adhered to the three core assumptions of MR [21]. In the genome-wide analysis, we performed variant selection and applied linkage disequilibrium (LD) clumping, with the following thresholds: P-value $< 5 \times 10^{-8}$, LD $r^2 = 0.001$, and genomic distance of 10,000 kbp.

To ensure the robustness of the results, we conducted sensitivity analyses, including tests for horizontal pleiotropy [29,30], heterogeneity analysis [31], and leave-one-out validation [32]. All analyses were performed in the R environment, primarily using the TwoSampleMR package (https://mrcieu.github.io/TwoSampleMR/) and the MR-PRESSO package (https://github.com/rondolab/MR-PRESSO) for data analysis and causal inference.

### 2.5. Conditional quantile-quantile (Q-Q) plots

The conditional Q-Q plot is employed as an intuitive visualization method for assessing polygenic enrichment patterns and genetic overlap across different phenotypes. Specifically, when the proportion of SNPs linked to one phenotype (e.g., IBD) exhibits a notable leftward shift as the *P*-values of another phenotype (e.g., BMD) decrease within the Q-Q plot, it provides compelling evidence for genetic enrichment and the presence of shared genetic variants between the two traits [33]. This shift represents more pronounced genetic signals concentrated in genomic regions common to both phenotypes.

To enhance the interpretation of results, *P*-values in all Q-Q plots were categorized into three intervals: $P < 0.10$, $P < 0.01$, and $P < 0.001$, reflecting the intensity and statistical significance of enrichment in a graduated scale. The Q-Q plots were generated using Python 3.11, along with the precimed/mixer package (https://github.com/precimed/mixer), ensuring both analytical efficiency and reproducibility. This methodology thus serves as an effective and powerful tool for investigating genetic associations between complex traits.

## 2.6. CondFDR/ConjFDR analysis

Through the systematic examination of GWAS data, the condFDR and conjFDR methodologies are shown to effectively identify shared genetic loci between complex traits and uncover their genetic commonalities [19]. The essence of the condFDR approach involves the use of SNP enrichment data from cross-trait analyses, facilitated by the construction of conditional Q-Q plots, where GWAS statistics for the principal phenotype are categorized and ordered based on the association levels of the secondary phenotype, thus enabling the calculation of condFDR values for each SNP. This methodology not only enhances the detection sensitivity for SNPs linked to the principal phenotype but also captures polygenic signals that may remain undetected in conventional GWAS studies. Building on condFDR findings, the conjFDR approach further computes joint FDR values for SNPs concurrently associated with both traits by reversing the roles of the principal and secondary phenotypes. The conjFDR value is defined as the maximum of the two condFDR values, thereby providing a more conservative estimate of shared genetic signals. This method does not rely on the overall Rg between traits when detecting shared genetic loci, thus enabling the identification of potential shared genetic mechanisms, even in cases where Rg is low [34]. By combining these two methodologies, the efficiency of identifying trait-associated SNPs is enhanced, while offering deeper insights into the shared genetic architecture and potential biological mechanisms underlying complex traits. This approach thus provides a solid theoretical foundation and technical support for the etiological investigation of complex traits and the identification of potential therapeutic targets.

The SNP loci selected by the conjFDR method require further functional annotation. This analysis was performed using the SNP2Gene module of the FUMA platform (https://fuma.ctglab.nl/) [35]. The main analysis parameters are as follows: SNP functional annotation was based on linkage LD data from the European population of the 1KGP; independent significant SNPs were selected with a condFDR < 0.05 and LD $r^2 \le 0.6$; among these, SNPs with LD $r^2 \le 0.1$ were defined as independent lead SNPs. For adjacent independent significant SNPs on the chromosome, if the physical distance between their LD block boundaries is < 250 kb, they were merged into the same genomic risk locus, with the SNP having the smallest P-value within the region being designated as the lead SNP.

The specific procedure is as follows: GWAS summary statistics were uploaded to the FUMA platform (https://fuma.ctglab.nl/) [35]; in the SNP2GENE module, the "MHC region" setting was selected to exclude the major histocompatibility complex (MHC) region to avoid interference from the complex LD structure in this region. The final annotation results include not only basic information based on genomic position but also integrate gene expression data and chromatin interaction information, providing a multidimensional reference for understanding the functional mechanisms of candidate genes.

## 2.7. Enrichment analysis

Functional enrichment analysis of the shared genes was carried out using the Sangerbox platform's enrichment tools (http://vip.sangerbox.com/) [36]. The gene set enrichment analysis in this process was conducted using the R package clusterProfiler (version 3.14.3) to identify significantly enriched biological pathways and functional modules. The key parameters for the analysis were set as follows: the gene set size was limited to between 5 and 5000 genes; the significance threshold was set at a p-value < 0.05, with FDR controlled below 0.1.

## 3. Results

### 3.1. Global Rg

Based on the results of the GNOVA analysis, notable negative Rgs were identified between IBD (including its subtypes, CD and UC) and BMD. Specifically, the Rg for the overall IBD and BMD was −0.1323 ($P$ = 2.4422e-05), suggesting that genetic variants linked to IBD exert a detrimental effect on bone density. Among the IBD subtypes, CD showed a more pronounced correlation with BMD, with an Rg of −0.128 ($P$ = 4.4617e-07), reflecting a strong negative association. Although somewhat weaker than CD, UC also displayed a statistically significant Rg with BMD (Rg = −0.1095, $P$ = 1.2856e-06) (Table 2).

Additional analysis identified a notable positive Rg between osteoporosis and IBD, with an Rg value of 0.0893 ($P$ = 0.0009). Among the IBD subtypes, a correlation coefficient of 0.0637 ($P$ = 0.0292) was observed between CD and osteoporosis, suggesting a statistically significant positive relationship. For the UC subtype, the Rg value with osteoporosis was 0.0959 ($P$ = 0.0006), further supporting a significant positive association (Table 2).

Moreover, the HDL analysis provided additional validation of these results, demonstrating highly consistent Rgs between IBD (including its subtypes CD and UC) and both BMD and osteoporosis, with comparable magnitudes of correlation coefficients (Table 2).

### 3.2. Local Rg

Local Rg analysis through LAVA identified specific chromosomal segments associated with IBD and BMD. Negative correlations were found in ten segments across chromosomes 1, 3, 6, 7, 8, 11, 12, and 20, while seven segments exhibiting positive correlations were observed on chromosomes 1, 3, 4, 5, 6, 7, and 12 (Fig 2A, Table 1 in S1 Tables). In the context of CD and BMD, fifteen correlated segments were detected, with six segments on chromosomes 3, 4, 5, and 9 showing positive correlations and nine segments on chromosomes 1, 7, 9, 12, and 20 displaying negative correlations (Fig 2B, Table 2 in S1 Tables]). For UC, fourteen chromosomal regions with local correlations were identified, among which nine segments on chromosomes 1, 6, 7, 8, and 9 exhibited negative correlations, and five segments on chromosomes 1, 3, 4, and 14 demonstrated positive correlations (Fig 2C, Table 3 in S1 Tables). These results highlight the intricate nature of local genetic associations between IBD (including its subtypes) and BMD, offering essential insights for further investigation of the genetic interaction mechanisms underlying these conditions.

The local Rg analysis between IBD and osteoporosis identified two chromosomal segments exhibiting significant negative correlations, both situated on chromosome 6 (Fig 2D, Table 4 in S1 Tables). In the context of CD, one positive correlation segment was detected on chromosome 8, alongside two negative correlation segments located on chromosomes 3 and 6, respectively (Fig 3E, Table 5 in S1 Tables). Furthermore, negative correlations with osteoporosis were also observed on chromosome 6 in the local analysis involving UC (Fig 2F, Table 6 in S1 Tables).

**Table 2. Genetic correlation of IBD (including CD and UC) with BMC and osteoporosis.**

| Trait1 | Trait2 | GNOVA-Rg | GNOVA-P | HDL-Rg | HDL-P |
|---|---|---|---|---|---|
| IBD | BMD | −0.1323 | 5.9139e-09 | −0.1311 | 4.12e-06 |
| CD | BMD | −0.128 | 4.4617e-07 | −0.1169 | 1.89e-05 |
| UC | BMD | −0.1095 | 1.2856e-06 | −0.1179 | 2.29e-04 |
| IBD | osteoporosis | 0.0893 | 0.0009 | 0.0713 | 1.77e-03 |
| CD | osteoporosis | 0.0637 | 0.0292 | 0.0631 | 1.36e-02 |
| UC | osteoporosis | 0.0959 | 0.0006 | 0.0814 | 2.96e-03 |

IBD, inflammatory bowel disease; CD, Crohn's disease; UC, ulcerative colitis; BMD, bone mineral density.

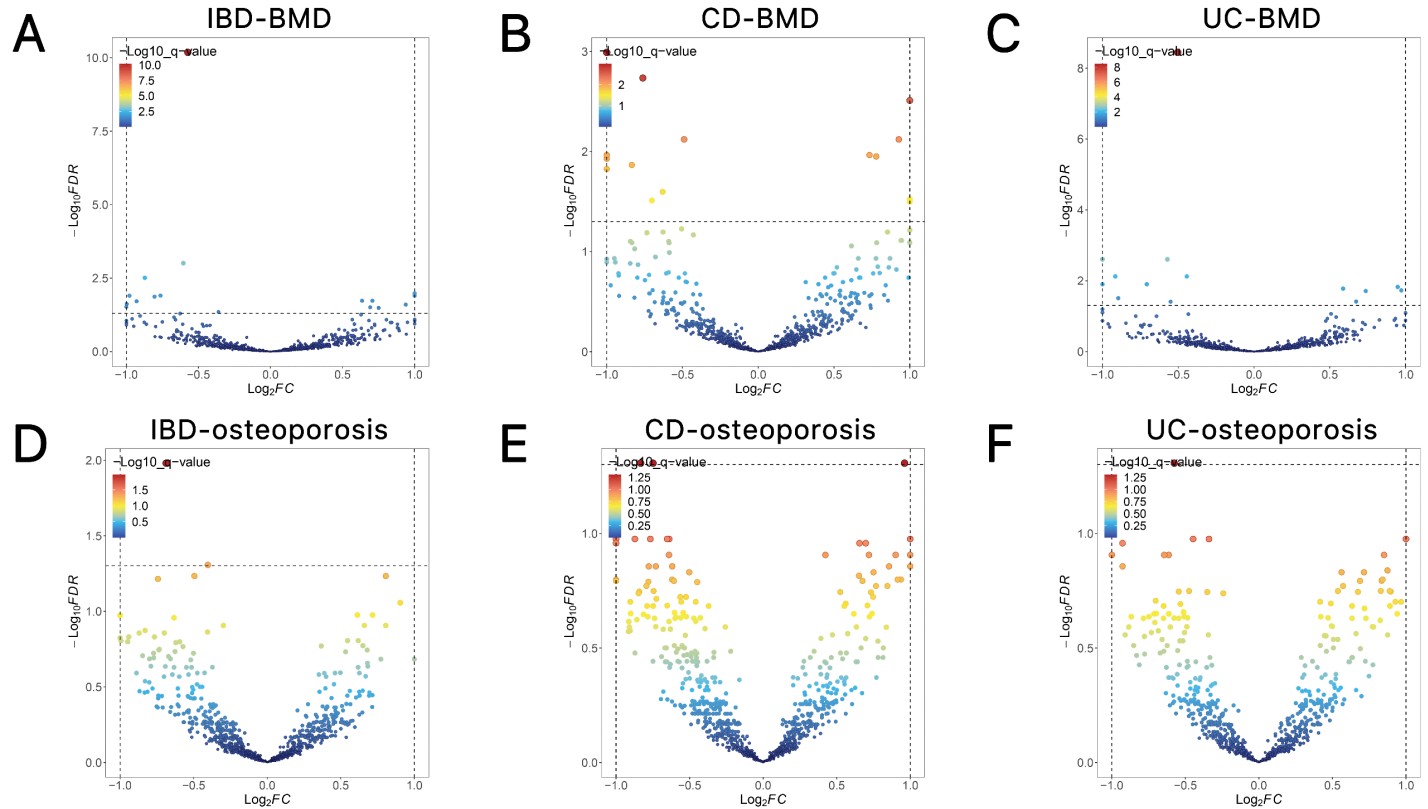

**Fig 2. LAVA analysis of IBD, BMD and osteoporosis.** The dashed line indicates the expected line with a correction P of 0.05. **(A)** Local genetic correlation between IBD and BMD. **(B)** Local genetic correlation between CD and BMD. **(C)** Local genetic correlation between UC and BMD. **(D)** Local genetic correlation between IBD and osteoporosis. **(E)** Local genetic correlation between CD and osteoporosis. **(F)** Local genetic correlation between UC and osteoporosis. IBD, inflammatory bowel disease; BMD, bone mineral density.

## 3.3. Mendelian randomization

This study employed bidirectional MR analysis to investigate the causal relationships between IBD, BMD, and osteoporosis. When IBD and its subtypes were considered as exposure variables and BMD as the outcome, IBD and UC were found to be negatively associated with BMD, a result consistent with previous MR studies [10]. When osteoporosis was treated as the outcome, a positive causal association was observed between IBD and CD [8–10]. No significant causal effects were detected in the reverse analysis.

Throughout all MR analyses, no evidence of horizontal pleiotropy was found, further confirming the validity and reliability of the selected instrumental variables. Additionally, all F-statistics exceeded the conventional threshold of 10, indicating that the instrumental variables were sufficiently strong and minimizing potential bias from weak instruments, thereby reinforcing the robustness of the causal estimates (Table 7 in S1 Tables). The leave-one-out sensitivity analysis demonstrated a consistent distribution of SNP effects, with no outlier variants detected.

## 3.4. ConjFDR analysis identifies shared genomic loci between two traits

The Q-Q plot results (Figs 3-4) reveal that as the association *P*-values for one trait (e.g., IBD) decrease, the corresponding association curve for another trait (e.g., BMD) undergoes a pronounced leftward shift, with the extent of deviation

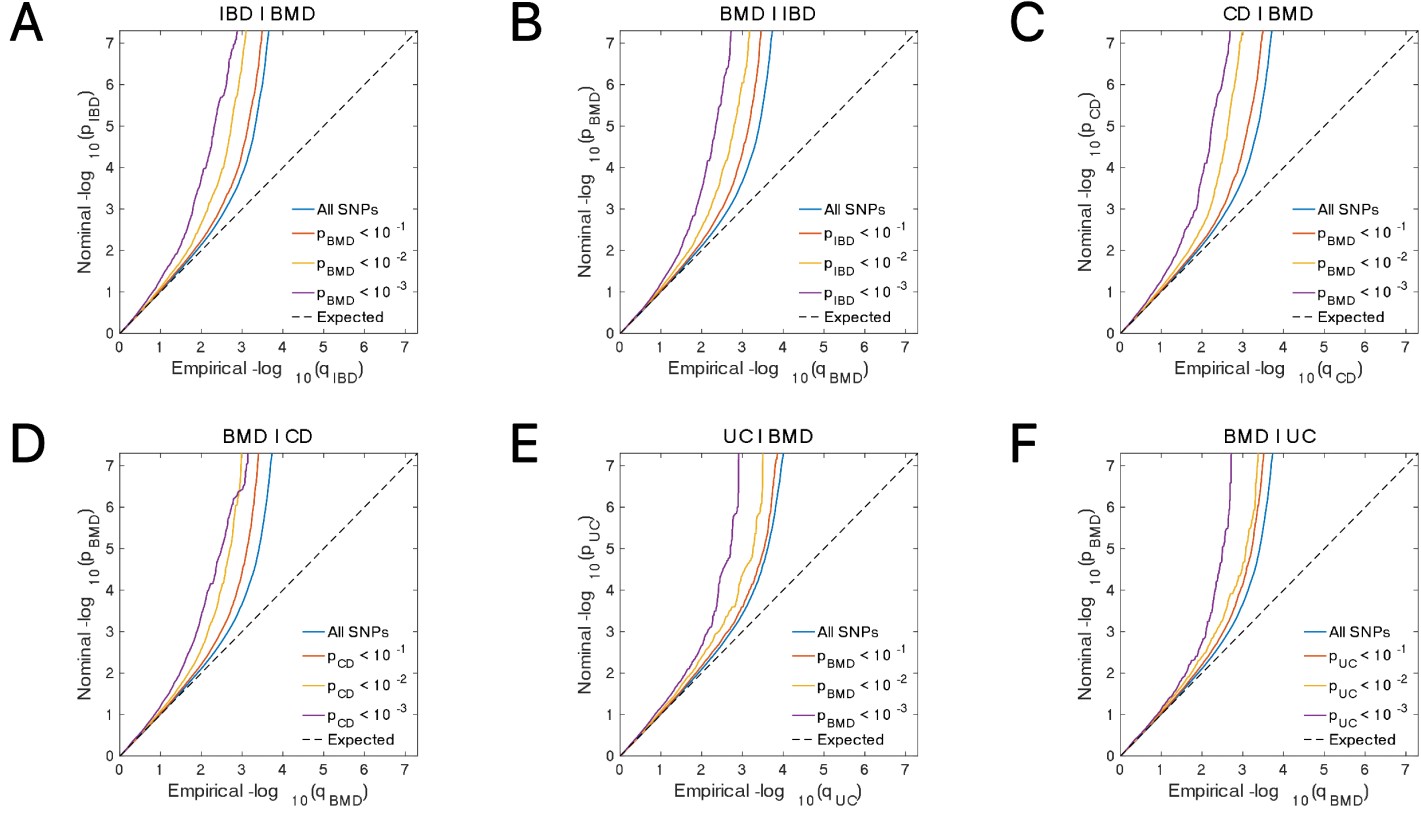

**Fig 3. Conditional quantile-quantile plot.** The dashed line indicates the expected line under the null hypothesis, and the deflection to the left indicates the degree of pleiotropic enrichment. **(A)** IBD-BMD. **(B)** BMD-IBD. **(C)** CD-BMD. **(D)** BMD-CD. **(E)** UC-BMD. **(F)** BMD-UC. IBD, inflammatory bowel disease; BMD, bone mineral density; CD, Crohn's disease; UC, ulcerative colitis.

steadily increasing. This observation points to a robust Rg between the two traits, implying the presence of shared genetic risk loci and, consequently, their genetic overlap.

The genetic overlap between BMD and IBD was systematically assessed through conjFDR analysis to pinpoint reliable shared genetic risk loci. A conjFDR value below 0.05 led to the identification of 58 shared genetic loci between these two traits, with 22 loci exhibiting consistent effect directions across both traits (Fig 5A, Table 8 in S1 Tables). Additional analysis focusing on the relationship between BMD and the IBD subtypes, CD and UC, revealed 45 and 8 shared loci, respectively. Of these, 14 loci demonstrated identical effect directions in both BMD and CD, while only a single locus showed concordant effect direction between BMD and UC (Fig 5B-C, Tables 9-10 in S1 Tables). The Gene Ontology (GO) analysis highlighted associations with immune system development, dephosphorylation, and regulation of binding. Additionally, KEGG pathway analysis revealed associations with cancer-related pathways and the Wnt signaling pathway (Fig 6A-B, Tables 11-12 in S1 Tables). These results provide substantial support and offer fresh insights into the genetic interplay and underlying mechanisms linking BMD and IBD, as well as their subtypes.

In the analysis of the genetic overlap between IBD and osteoporosis, seven shared loci were detected, five of which displayed consistent effect directions across both conditions (Fig 7A, Table 13 in S1 Tables). A similar pattern was observed in the comparison between CD and osteoporosis, where seven shared loci were identified, with five demonstrating concordant effect directions (Fig 7B, Table 14 in S1 Tables). In the case of UC, four shared loci were found, all of which exhibited consistent effect directions between the two traits (Fig 7C, Table 15 in S1 Tables). Enrichment analysis

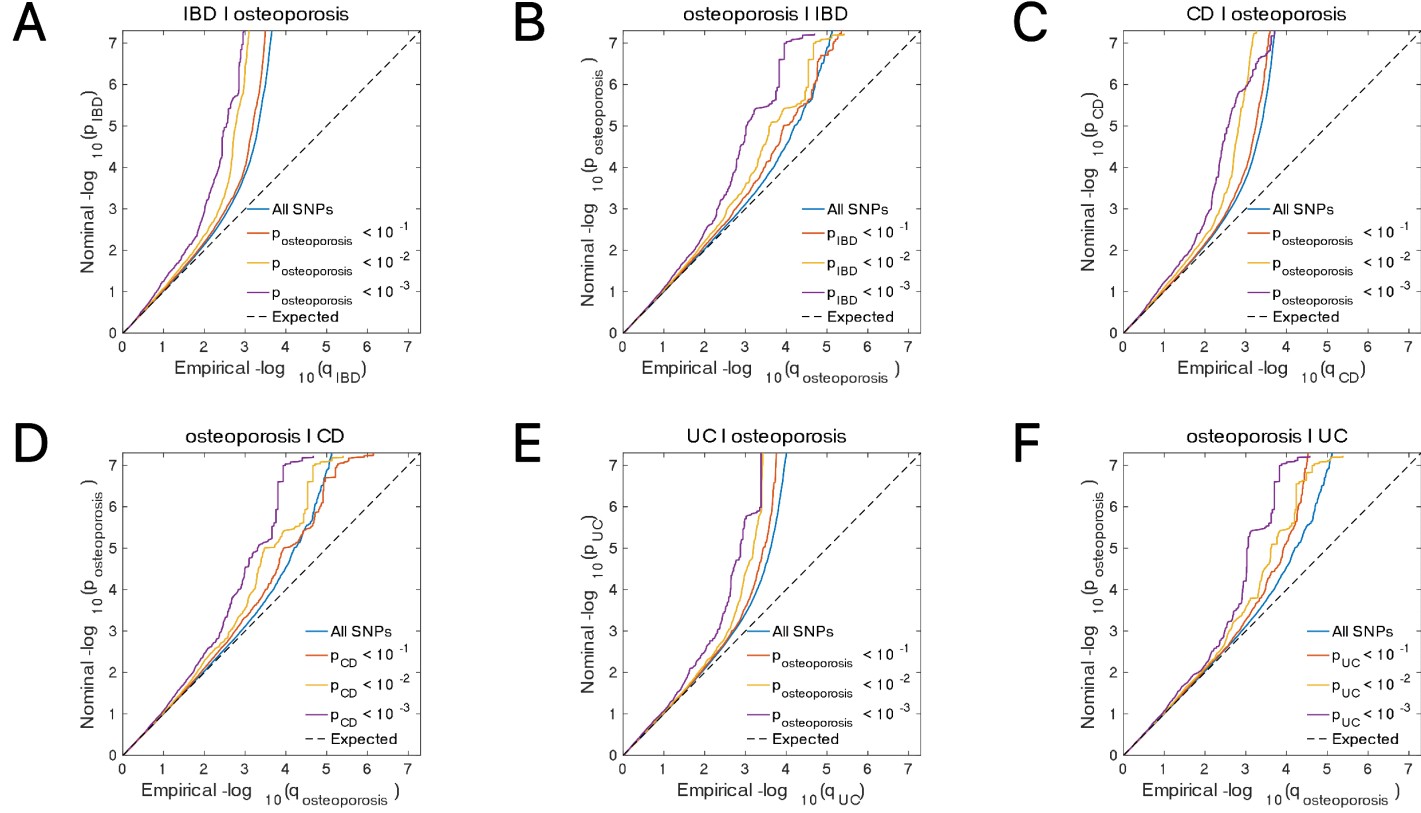

**Fig 4. Conditional quantile-quantile plot.** The dashed line indicates the expected line under the null hypothesis, and the deflection to the left indicates the degree of pleiotropic enrichment. **(A)** IBD-osteoporosis. **(B)** osteoporosis-IBD. **(C)** CD-osteoporosis. **(D)** osteoporosis-CD. **(E)** UC-osteoporosis. **(F)** osteoporosis-UC. IBD, inflammatory bowel disease; CD, Crohn's disease; UC, ulcerative colitis.

of the shared loci between IBD and osteoporosis revealed that the GO and KEGG pathways were markedly enriched in processes related to animal organ development and the Wnt signaling pathway, respectively (Fig 8A-B, Tables 16-17 in S1 Tables).

The results of the analysis on BMD and osteoporosis were subsequently structured, uncovering four shared genes in IBD: ZBTB40, RP1-135L22.1, RSPO3, and RP11-103J8.1 (Fig 9A). In the CD analysis, three common genes were identified, including RP1-135L22.1, RSPO3, and RP11-103J8.1 (Fig 9B). For UC, two overlapping genes were detected: ZBTB40 and RP1-135L22.1 (Fig 9C).

## 4. Discussion

A series of in-depth genetic analyses were conducted to explore the shared genetic foundation between IBD, its subtypes (CD and UC), and BMD/osteoporosis. By employing various analytical methodologies, including GNOVA, HDL, LAVA, and conjFDR, the genetic overlap between IBD and its subtypes with BMD and osteoporosis was more comprehensively defined, revealing a set of shared genes that likely contribute to both conditions. Specifically, the global Rg was characterized, and the local genetic influences within specific chromosomal regions were further examined, leading to the identification of several robust risk genes that impact both traits. These results not only enhanced the understanding of the genetic interplay between IBD, BMD, and osteoporosis but also offered substantial theoretical and empirical support for advancing the elucidation of the underlying common genetic mechanisms.

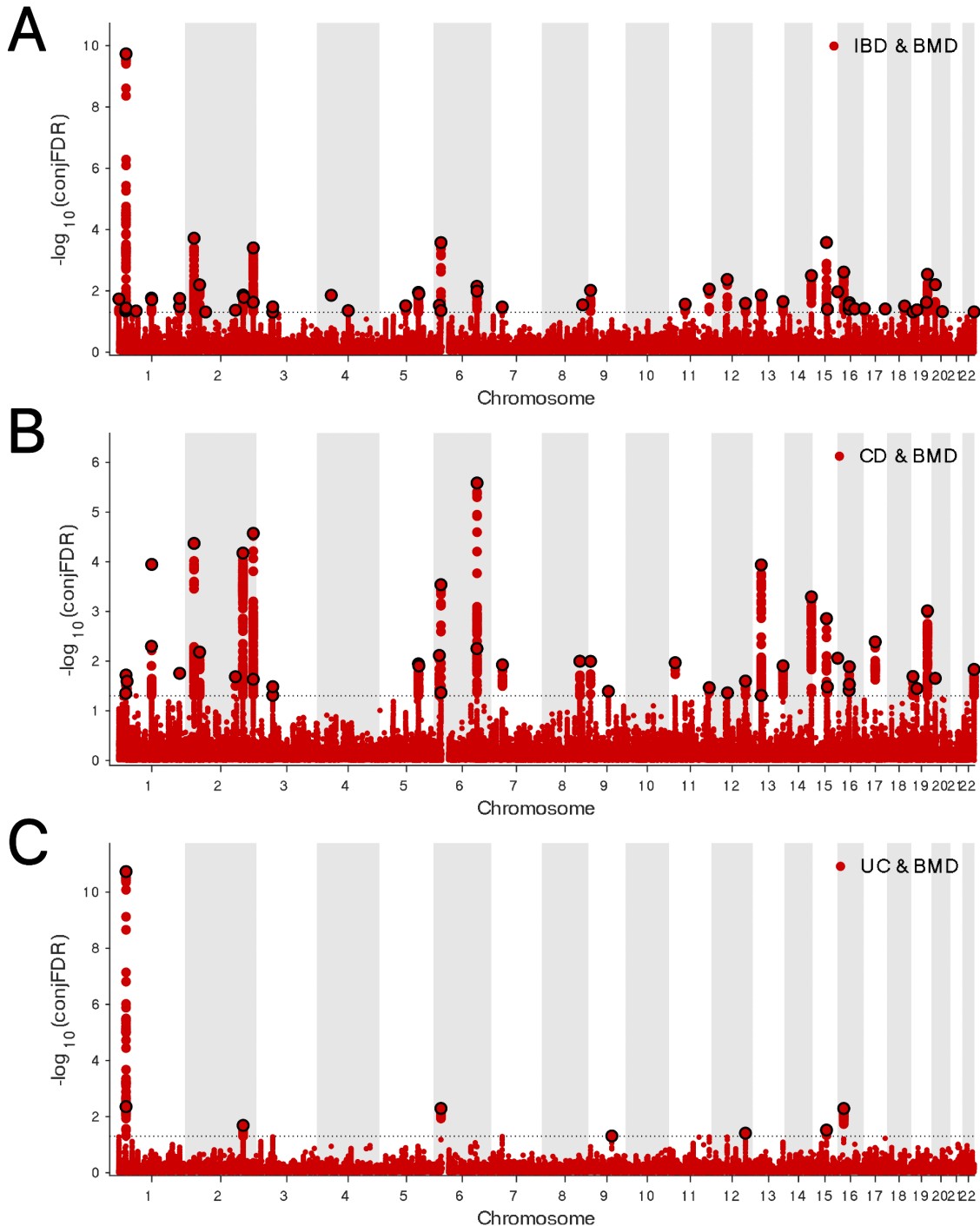

**Fig 5. (A) ConjFDR Manhattan plot of IBD and BMD. (B)** ConjFDR Manhattan plot of CD and BMD. **(C)** ConjFDR Manhattan plot of UC and BMD. The shared risk loci between BMD and IBD, CD and UC were marked. The statistically significant causality is defined to be conjFDR<0.05. BMD, bone mineral density; IBD, inflammatory bowel disease;CD, Crohn's disease; UC, ulcerative colitis.

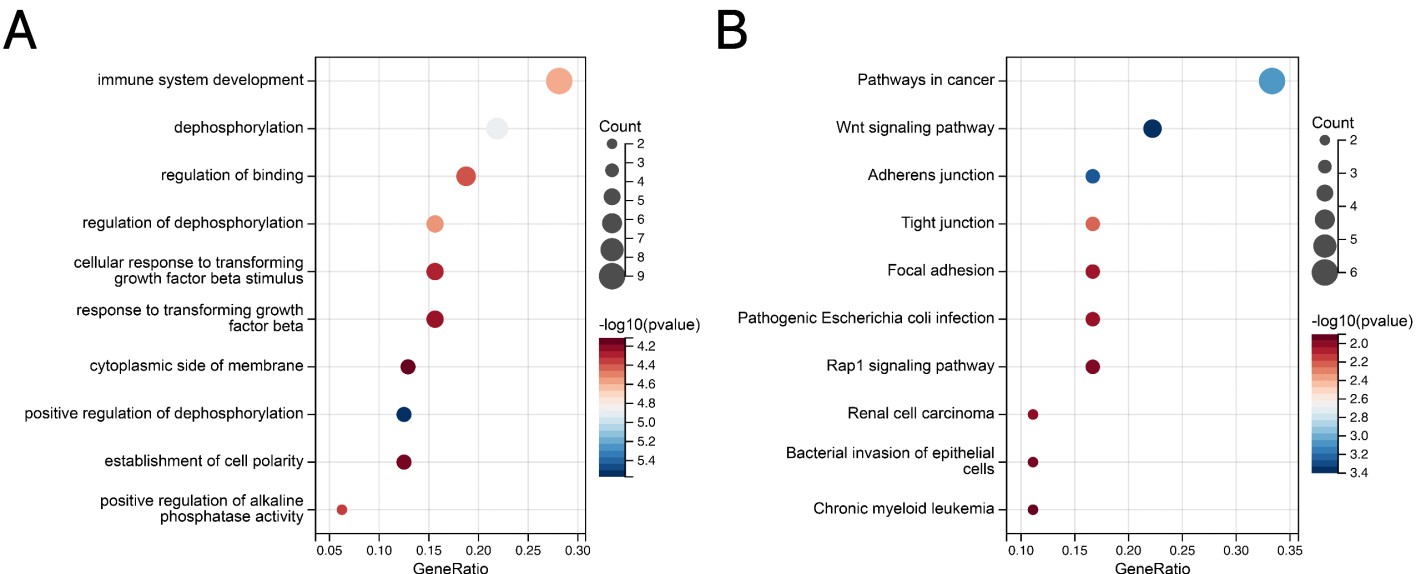

**Fig 6. (A) GO enrichment analysis of mapped genes between IBD and BMD. (B)** KEGG enrichment analysis of mapped genes between IBD and BMD. BMD, bone mineral density; IBD, inflammatory bowel disease.

The genome-wide Rg analysis revealed substantial genetic associations between IBD, including its subtypes (CD and UC), and both BMD and osteoporosis, aligning with prior studies. For example, a survey conducted on Polish IBD patients reported that osteoporosis prevalence was 48.6% in CD patients and 33.3% in UC patients. Furthermore, IBD patients diagnosed with osteoporosis exhibited markedly lower levels of physical activity ($P=0.0335$) [37]. A separate multinational prospective investigation documented that 12%−22% of IBD patients had asymptomatic vertebral fractures, with prevalence rates of 19.6% (44/224) in Canada, 21.8% (34/156) in Germany, and 12.2% (22/179) in Israel [38,39]. Research by D. Leslie et al. also corroborated these findings, demonstrating that IBD patients showed reduced BMD and a markedly elevated risk of osteoporosis [40]. Additionally, analyses conducted by C. Noble et al. using both univariate and multivariate approaches indicated that a low body mass index (<18.5) was strongly associated with osteoporosis ($P=0.021$, OR: 5.83, CI: 1.31–25.94) [41]. A study from Sri Lanka involving 444 participants (case-control ratio of 1:3) further revealed that DXA bone density scans indicated a markedly higher overall incidence of osteoporosis in IBD patients compared to controls (13.5% vs. 4.5%, $P=0.001$) [42]. Together, these studies establish a clear association between IBD and reduced bone density and osteoporosis, thus offering critical insights into their underlying pathological mechanisms. However, certain studies have reported conflicting results. Early investigations employing multivariate linear and logistic regression models found no significant differences in osteoporosis prevalence (based on T-scores) between CD and UC patients [43]. Additionally, research by Schoon et al. using dual-energy X-ray absorptiometry to assess total body, spine, and hip BMD found no significant differences between CD and UC patients, nor was there any notable decline in BMD in newly diagnosed IBD patients [44]. To address these inconsistencies, a systematic genetic analysis of the association between IBD and BMD/osteoporosis was conducted, with careful consideration of potential confounding factors. By integrating a variety of statistical techniques, the genetic association between these conditions was further validated, ensuring the robustness and reliability of the findings. This study offers new genetic insights and provides substantial support for elucidating the comorbid mechanisms underlying IBD and BMD/osteoporosis.

The conjFDR method successfully identifies multiple significant genetic variation loci distributed across different chromosomal regions. A comprehensive analysis of the local genetic relationship between these two diseases suggests that

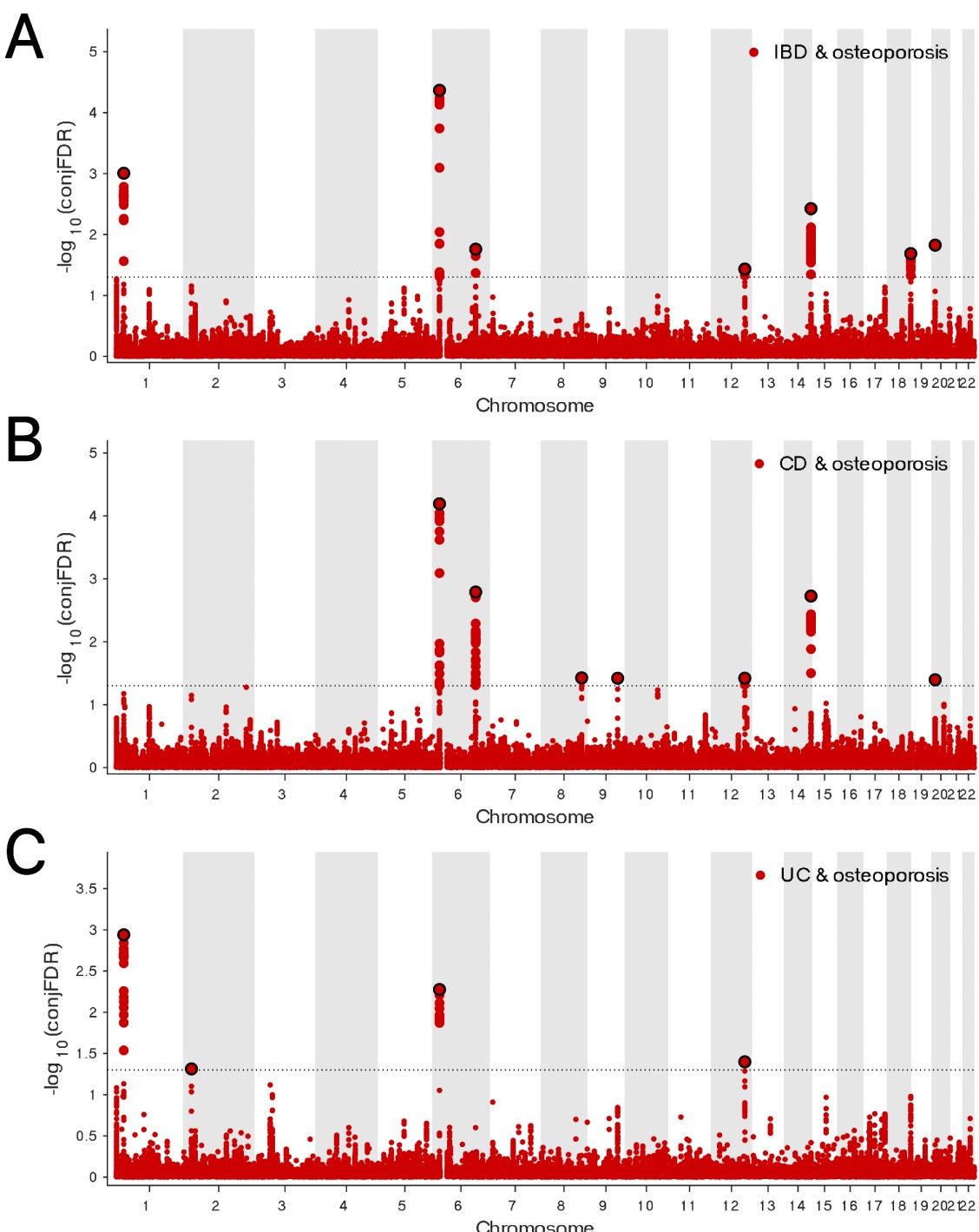

**Fig 7. (A) ConjFDR Manhattan plot of IBD and osteoporosis. (B)** ConjFDR Manhattan plot of CD and osteoporosis. **(C)** ConjFDR Manhattan plot of UC and osteoporosis. The shared risk loci between osteoporosis and IBD, CD and UC were marked. The statistically significant causality is defined to be conjFDR < 0.05. IBD, inflammatory bowel disease; CD, Crohn's disease; UC, ulcerative colitis.

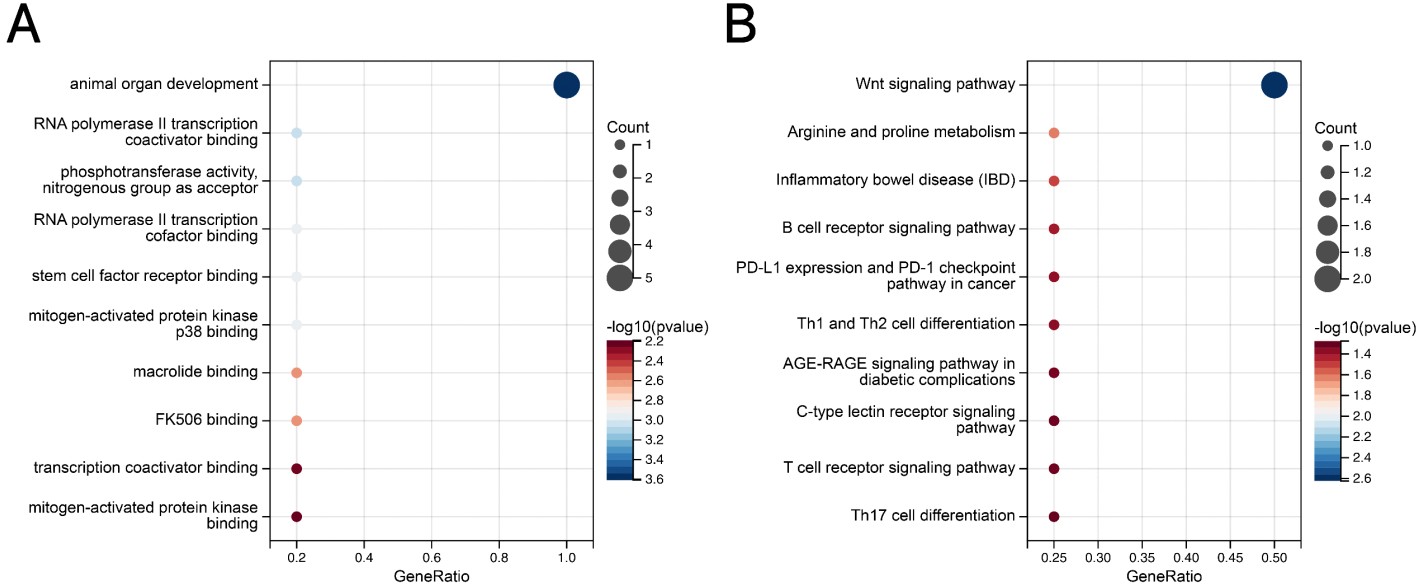

**Fig 8. (A) GO enrichment analysis of mapped genes between IBD and osteoporosis. (B)** KEGG enrichment analysis of mapped genes between IBD and osteoporosis. IBD, inflammatory bowel disease.

certain chromosomal segments may play a crucial role in their shared pathogenic mechanisms, potentially regulated by functional genes located within these segments, such as ZBTB40 and RSPO3. Zinc Finger and BTB Domain Containing 40, a transcription factor, serves a pivotal function in regulating bone metabolism and inflammation. GWAS has revealed that genetic variants of ZBTB40 are strongly associated with BMD, with specific loci, such as rs6426749, being closely correlated with bone loss and fracture susceptibility [45,46]. On a molecular level, ZBTB40 modulates bone metabolism by influencing osteoblast activity and regulating pathways involved in bone formation [47]. In the context of IBD, studies have documented substantial changes in ZBTB40 expression within the intestinal tissues of IBD patients [48], indicating its potential role in the pathogenesis of IBD through modulation of immune cell functions and cytokine production [49]. ZBTB40's involvement is particularly pronounced in IBD-related osteoporosis. Clinical evidence suggests that IBD patients carrying certain ZBTB40 variants exhibit an increased risk of fractures [50], which may be attributed to the factor's dual regulatory influence on both bone metabolism and inflammatory processes. Furthermore, ZBTB40 is proposed to regulate BMD by affecting vitamin D metabolism and calcium ion homeostasis [51], thus offering novel mechanistic insights into secondary osteoporosis observed in IBD patients. Current findings suggest that ZBTB40 may be a key molecular link between IBD, BMD regulation, and osteoporosis development, thus presenting substantial clinical translational potential [52]. R-spondin 3 (RSPO3), a pivotal modulator of the Wnt signaling pathway, serves an essential function in bone metabolism and immune responses. Studies have shown that RSPO3 directly impacts BMD by enhancing Wnt/β-catenin pathway activity, thus promoting osteoblast differentiation and proliferation [53]. In IBD, RSPO3 expression levels show considerable alterations, which are closely associated with the function of the intestinal epithelial barrier and mucosal repair [54]. Multiple RSPO3 genetic variants have been identified through GWAS, with SNPs such as rs13204965 showing a significant relationship with BMD, potentially increasing the risk of osteoporosis [45]. In IBD patients, abnormal RSPO3 expression may exacerbate bone loss indirectly by influencing the intestinal microenvironment and modulating inflammatory cytokine production [55]. Mechanistically, RSPO3 regulates genes associated with bone metabolism by binding to LGR4/5 receptors, thereby amplifying and prolonging Wnt signal transmission [56]. Clinical studies have highlighted a significant correlation between RSPO3 expression and BMD in IBD patients, offering new insights into the pathogenesis

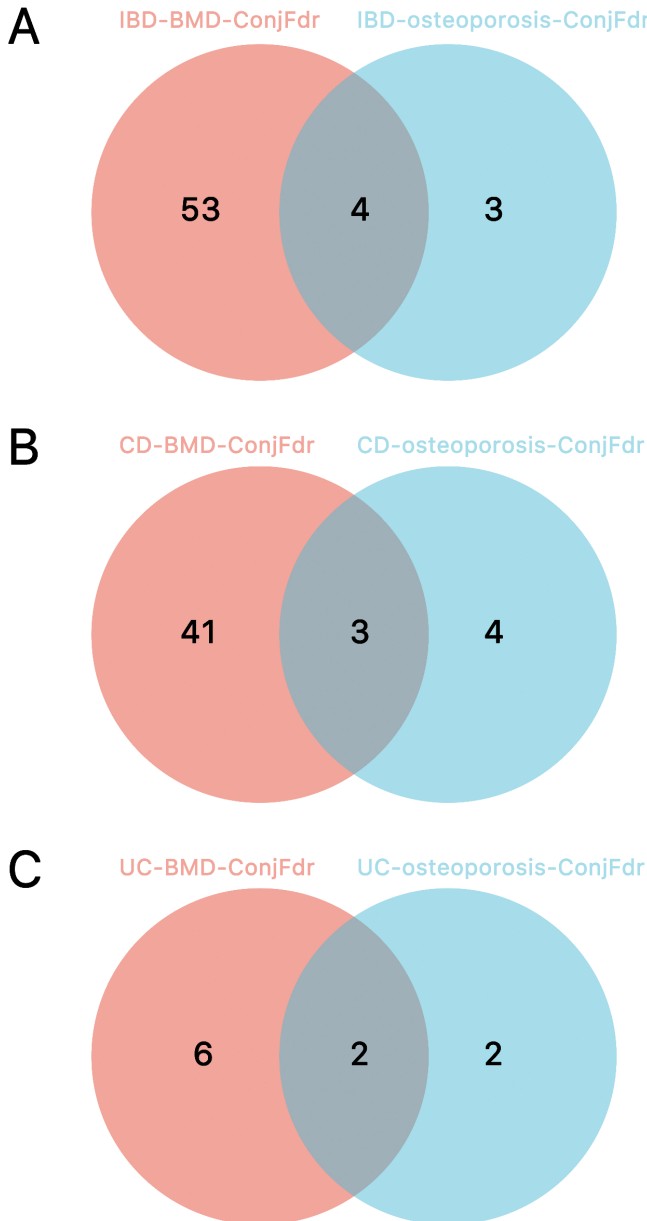

**Fig 9. The results of the intersection genes of the two analyses. (A)** IBD. **(B)** CD. **(C)** UC.

of IBD-related osteoporosis [57]. Recent findings further suggest that RSPO3 could serve as a critical intermediary in the development of IBD-related osteoporosis by modulating immune cell functions and the expression of inflammatory factors [58]. The role of non-coding RNAs (RP1-135L22.1 and RP11-103J8.1) in IBD and BMD/osteoporosis remains underexplored, indicating the need for further investigation in future studies.

The Wnt signaling pathway, which was identified through enrichment analysis, warrants further investigation. As a highly conserved mechanism of signal transduction, this pathway is integral to the regulation of bone metabolism and the maintenance of intestinal homeostasis. It has been demonstrated that the canonical Wnt/β-catenin signaling pathway

directly influences BMD by modulating osteoblast differentiation and proliferation [59]. In the context of IBD, the abnormal activation of the Wnt signaling pathway is strongly linked to dysfunction of the intestinal epithelial barrier and the onset of inflammatory responses [60]. Of particular note, in patients with inflammatory chronic disease, significant changes in the expression of essential Wnt pathway components (e.g., LRP5/6 and β-catenin) have been reported, which not only impair the repair of the intestinal mucosa but may also exacerbate bone loss via immunomodulatory mechanisms [61]. Investigations into the molecular mechanisms have shown that inflammatory cytokines, such as TNF-α and IL-1β, can inhibit the activity of the Wnt signaling pathway, thus leading to diminished osteoblast differentiation and an increase in bone resorption [62]. Furthermore, the Wnt signaling pathway functions in osteoclast activation through modulation of the RANKL/OPG system, a process that is essential in the pathogenesis of IBD-associated osteoporosis [63]. Clinical data have revealed a notable association between the elevated levels of Wnt inhibitors, such as DKK1 and SOST, and the reduced BMD observed in IBD patients [64]. Notably, this pathway also contributes to the regulation of intestinal stem cell self-renewal and differentiation, processes that are vital for maintaining the integrity of the intestinal epithelium and mitigating the progression of IBD [65]. Recent studies have suggested that targeting the Wnt signaling pathway could offer novel therapeutic approaches for treating IBD-related osteoporosis [66].

This study systematically examined the link between IBD and BMD/osteoporosis through the lenses of Rg and shared loci. Various methodologies were utilized to uncover genetic associations at both the genome-wide and SNP levels, thereby broadening the existing knowledge. Nevertheless, certain limitations warrant attention: the influence of LD could not be entirely excluded, potential sample overlap may exist, and the data were exclusively sourced from European populations, restricting cross-ethnic generalizability. The GWAS datasets used in this study were not stratified by sex, age, or disease stage, limiting our ability to explore sex-, age-, and stage-specific genetic susceptibility to IBD, BMD, and osteoporosis comorbidity. Future analyses will address this limitation if such data become available. Furthermore, the conclusions primarily relied on computational simulations, lacking validation through real-world population data. Future investigations could address these issues by incorporating GWAS data from more diverse populations and performing trans-ethnic analyses, which would enhance the understanding of the connection between IBD and bone health.

## 5. Conclusion

In conclusion, this study uncovered significant Rgs between IBD and BMD/osteoporosis, suggesting a certain degree of genetic overlap between these disorders. The results not only offer new genetic evidence for clinical applications but also enhance the comprehension of their intricately connected genetic framework. Furthermore, several shared genetic loci that may serve pivotal functions in the interaction between IBD and BMD/osteoporosis were successfully identified and validated. These findings provide critical insights for future investigations into the common biological mechanisms underpinning both conditions and establish a solid scientific foundation for the development of potential therapeutic strategies.

## Supporting information

**S1 Tables. Comprehensive summary of supplementary results.** This Excel file provides the complete corresponding analysis results, organized into 17 worksheets [Tables 1–17], each labeled to match the in-text references. (XLSX)

**S1 File. STROBE-MR-checklist-fillable.** (DOCX)

## Acknowledgments

The authors thank Bullet Edits Limited for the linguistic editing and proofreading of the manuscript.

## Author contributions

**Conceptualization:** Mengting Qin.

**Data curation:** Mengting Qin.

**Formal analysis:** Mengting Qin.

**Funding acquisition:** Xinhong Liu.

**Methodology:** Xinhong Liu, Ruyun Cai.

**Supervision:** Qinghua Luo.

**Validation:** Xinhong Liu.

**Writing – original draft:** Mengting Qin, Xinhong Liu, Qinghua Luo.

**Writing – review & editing:** Ruyun Cai.

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
