## [Decision Letter · Decision Letter 0]

1 Oct 2025

Dear Dr. Qin,

Thank you for submitting your manuscript to PLOS ONE. After careful consideration, we feel that it has merit but does not fully meet PLOS ONE’s publication criteria as it currently stands. Therefore, we invite you to submit a revised version of the manuscript that addresses the points raised during the review process.

**ACADEMIC EDITOR: ** Reviewers have provided feedback. Authors need to pay close attention and respond to the comments, making corrections to their manuscript as necessary. As noted by some of the reviewers, there is a need to strengthen methodological rigour and provide more detail about the data utilised.

Is there a reason the authors did not perform Mendelian randomisation to assess potential causal associations? Yes, I see a reference to previous studies, but could they have assessed this to confirm or contextualise their findings? It would probably add depth and context to their study.

We look forward to receiving your revised manuscript.

Kind regards,

Emmanuel O Adewuyi, BPharm, MPH, PhD

Academic Editor

PLOS ONE

Journal Requirements:

“This study was supported by the Jiangxi Provincial Natural Science Foundation project (20232BAB206154) and the University-Level Research Project of Zhejiang Chinese Medical University (2022FSYYZQ21).”

Reviewers' comments:

Reviewer's Responses to Questions

**Comments to the Author**

1. Is the manuscript technically sound, and do the data support the conclusions?

Reviewer #1: Yes

Reviewer #2: Yes

Reviewer #3: Partly

Reviewer #4: Yes

Reviewer #5: Partly

2. Has the statistical analysis been performed appropriately and rigorously?

Reviewer #1: Yes

Reviewer #2: No

Reviewer #3: Yes

Reviewer #4: Yes

Reviewer #5: Yes

3. Have the authors made all data underlying the findings in their manuscript fully available?

Reviewer #1: Yes

Reviewer #2: No

Reviewer #3: Yes

Reviewer #4: Yes

Reviewer #5: No

4. Is the manuscript presented in an intelligible fashion and written in standard English?

Reviewer #1: Yes

Reviewer #2: Yes

Reviewer #3: Yes

Reviewer #4: Yes

Reviewer #5: Yes

Reviewer #1:

In the methods:

-The total number of participants in the study did not specify the number of males and females or the participants age distribution to identify the more affected sex and age.

- As the IBD (Crohn’s disease (CD) and ulcerative colitis (UC)) is mentioned in lines 32 and 33, it is expected the immune response genes are involved, but genes that govern the immune response, like HLA genes, or genes related to the absorption and metabolism of calcium, potassium, phosphorus, etc., and genes related to vitamin D, such as VDR, etc., are not analyzed in the study, which may have an important impact on bone density.

-It will be a significant addition if further genes are included and their role in bone density is elucidated in a large number of populations and categorized according to their sex and age.

Reviewer #2: Abstract

Clarify “biologically significant” loci. Does this mean functional annotation, pathway relevance, or previous association?

Add the total number of GWAS datasets or total sample size used for clarity.

Introduction

Update references to include more recent global epidemiological data where possible.

The paragraph beginning on line 51 is overly dense. Consider splitting into two, and briefly introduce each analytic tool in one sentence before elaborating.

Methods and Materials

Specify the exact sample size for each GWAS used and any data pre-processing or quality control performed (SNP filtering, ancestry exclusion) (Line 84–91).

State the specific genomic resolution used in the LAVA (e.g., 1Mb blocks? predefined regions?). Also, clarify if any Bonferroni or FDR correction was applied in interpreting local correlations (Line 120–130).

Clarify whether the FUMA gene annotations were only positional or included eQTL, chromatin interactions, etc.

Results

Enrichment Results (lines 216–231): Clarify how significance was defined in GO and KEGG enrichment (e.g., FDR < 0.05?).

Were any of the identified shared loci previously known in either disease, or are they novel findings?

References

Ensure all URLs (e.g., for software tools like FUMA, Sangerbox) are active and appropriately cited. (Line 350-351) “All GWAS data and statistical software utilized in this study were publicly available (accessible via the following URLs)…”

Reviewer #3: The manuscript by Qin et al. has explored the association between inflammatory bowel disease (IBD) and bone mineral density (BMD) and osteoporosis with extensive statistical tests, and concluded that a notable genetic correlation exists between IBD and BMD/osteoporosis.

The introduction was well written with clear explanation of different statistical methods for GWAS testing and Rg analysis. And the discussion went to great lengths to review the genes/pathways and their potential roles in the development of IBD/BMD/osteoporosis.

However, there are a few questions regarding the design and interpretation of the work, hope the authors could address before considered for publication:

1. Since the study was solely based on the statistical testing and computational simulation of the GWAS data, could the authors provide more details and characterization of the data? I.e. the sample size, age group, stage of IBD/osteoporosis etc., as well as the features that were selected for statistical testing, which would affect the interpretation of correlation scores.

2. Have the authors taken into account possible confounding factors that could cause spurious correlations? I.e. both osteoporosis and IBD (especially CD) has age-related manifestation, which might require additional testing or randomization to rule out the cofounding effect.

3. To further validate the findings, could the author include negative controls into the testings (i.e. using unrelated/negative samples, or check against unrelated symptoms)?

Reviewer #4: In this paper, the authors utilized data from GWAS on IBD, its subtypes, BMD, and osteoporosis to conduct comprehensive analyses with focuses on genetic correlations and shared genetic loci. In addition, local genetic patterns were examined using local variant association analysis. As a result, a notable genetic correlation was detected between IBD, its subtypes, and BMD/osteoporosis. Overall, the paper is well-written. Below are my comments on the paper.

• In Section 3.2, the authors mentioned LAVA identified specific chromosomal segments associated with IBD and BMD. I would recommend the authors to clearly indicate how the chromosomal segments are determined either in section 2.3 or in section 3.2.

• The resolutions of the figures in the manuscript needs improvement. For instance, it is hardly to see any numbers and labels on the horizontal and vertical axes in Figure 2.

Reviewer #5: The authors conducted a series of genetic analyses to investigate the shared genetic contribution to IBD and bone mineral density (BMD)/osteoporosis. Through integration of their findings, they identified four genes that may be regulated by variants associated with both IBD and BMD/osteoporosis. The approach is interesting and potentially valuable for advancing our understanding of the shared genetic predisposition underlying these complex phenotypes.

However, the manuscript lacks in-depth interpretation and integration of the results obtained from the different analytical strategies. The use of global Rg, local Rg, and conjFDR is described, but the interplay between these methods—and whether their results corroborate one another—is not sufficiently explored. From this reviewer’s perspective, the results are primarily reported in a descriptive manner, with limited synthesis or biological insight. In addition, the manuscript suffers from insufficient methodological transparency, which limits reproducibility. For these reasons, the analyses require deeper investigation and clearer integration before the work can be considered for publication.

Major comments:

1. The Methods section requires greater detail to ensure reproducibility:

a. It is not clear which LD matrix was used for the HDL and LAVA tools. Please specify, as these methods are sensitive to the structure of the reference LD matrix. Furthermore, was any filtering applied to the reference genetic data to ensure homogeneous European ancestry before calculating the LD matrix?

b. For the Local Analysis of Variant Association, the manuscript vaguely refers to “user-defined genomic regions.” While the supplementary materials (Tables S1–S6) indicate that the analysis was performed on a set of predefined, non-overlapping genomic blocks with specific coordinates, the Methods section should explicitly describe how these regions were defined.

c. The procedure for enrichment analysis is not described in the Methods. Some information appears in the Results (lines 216–219), but this should be moved to the Methods and expanded.

d. The parameters used in FUMA for the SNP2gene module are not reported. Were specific variant or gene annotations applied in this step?

2. In the local Rg and ConjFDR analyses, the results obtained for the IBD subtypes (UC and CD) appear to largely recapitulate those observed when considering IBD as the overall phenotype. The authors should further discuss which loci show genetic correlation only at the subtype level versus those consistent across the broader IBD phenotype, and clarify which loci specifically overlap with the IBD-wide analysis. In addition, the overlap of genetic correlation results between BMD and osteoporosis should be addressed.

3. Concordance between methods

Do the ConjFDR findings corroborate those of the local Rg analysis? Specifically, do the loci identified in local Rg include the variants highlighted by ConjFDR?

4. What did the FUMA analysis reveal? Were the identified variants located in regulatory regions, coding regions, or both?

5. The results of the enrichment analysis should be presented more clearly, ideally in supplementary tables. For example, which genes identified through ConjFDR and FUMA analyses are included within the enriched pathways?

6. Why did the authors restrict their focus to variants and genes common to both BMD and osteoporosis? Were these shared genes enriched in specific pathways? How do the other identified genes (those not shared between BMD and osteoporosis) behave—do they also map to relevant pathways?

Minor comments:

1. In Figure 1, the workflow gives the misleading impression that the output of each analysis serves as the input for the subsequent one. I recommend revising the figure to clearly indicate that the input for all three analyses is the GWAS summary statistics.

**Do you want your identity to be public for this peer review?**  For information about this choice, including consent withdrawal, please see our Privacy Policy

Reviewer #1: **Yes: ** Luma Hassan Alwan Al Obaidi

Reviewer #2: No

Reviewer #3: No

Reviewer #4: No

Reviewer #5: No

---

## [Author Response · Author response to Decision Letter 1]

5 Oct 2025

RESPONSE LETTER

Oct. 05, 2025

Dear Prof. Emmanuel O Adewuyi, BPharm,

Thank you for reviewing our manuscript (Manuscript ID: PONE-D-25-31485; Manuscript title: A Multilayered Genetic Structure Analysis between Inflammatory Bowel Disease and Bone Density/Osteoporosis) and for your decision of further consideration. Now we have revised and checked the manuscript carefully, according to the editor’s and reviewer’s comments.

We submit here the revised paper. The changes and explanations regarding the revisions of our paper are listed below this letter.

We have tried our best to address all the concerns raised by the editor and reviewer. We hope that, with these modifications and improvements based on the editor’s and reviewer’s comments, our manuscript would be acceptable for publication in your journal.

Thank you again!

With best wishes,

Mengting Qin

Department of Anorectal Surgery, Zhejiang Chinese Medical University Affiliated Jiaxing TCM Hospital, Jiaxing, China

Email: qinmt0907@163.com

Reply to Editor

Thank you for submitting your manuscript to PLOS ONE. After careful consideration, we feel that it has merit but does not fully meet PLOS ONE’s publication criteria as it currently stands. Therefore, we invite you to submit a revised version of the manuscript that addresses the points raised during the review process.

Response:

Thank you for your feedback and for considering our manuscript.

We have carefully reviewed the reviewer's comments and appreciate their valuable feedback. We acknowledge the importance of enhancing the clarity and understanding of our work, and we have already implemented the suggested modifications to address these concerns (please see the point-by-point response below for details).

Thank you again!

ACADEMIC EDITOR:

#1:Reviewers have provided feedback. Authors need to pay close attention and respond to the comments, making corrections to their manuscript as necessary. As noted by some of the reviewers, there is a need to strengthen methodological rigour and provide more detail about the data utilised.

Response:

We sincerely thank the reviewer for the constructive feedback. In response to the comments, we have revised the manuscript to strengthen the methodological clarity and provided additional details regarding the data utilized. The specific changes are summarized below.

#2:Is there a reason the authors did not perform Mendelian randomisation to assess potential causal associations? Yes, I see a reference to previous studies, but could they have assessed this to confirm or contextualise their findings? It would probably add depth and context to their study.

Response:

We sincerely thank the academic editor for this valuable suggestion. We have taken it into account and included a Mendelian randomisation analysis in the revised manuscript (see sections 2.4, lines 161-171, and 3.3, lines 264-276). The results of this analysis are consistent with previous studies, and we believe that this addition has enhanced the depth and context of our findings. We greatly appreciate the editor’s contribution to improving the manuscript.

Reply to Reviewer 1

In the methods:

#1: The total number of participants in the study did not specify the number of males and females or the participants age distribution to identify the more affected sex and age.

Response:

We sincerely thank the reviewer for the valuable suggestion. We have now provided additional details regarding the cohort information in the revised manuscript, including Phenotypes, Phenotypic code, Cases/Controls, and Ancestry (see Table 1, lines 695-697). We acknowledge that the current study did not specify the number of males and females or the participants' age distribution. This is due to the lack of sex- and age-stratified data in our current dataset, which limited our ability to explore sex- and age-specific genetic susceptibility to IBD, BMD, and osteoporosis comorbidity. Therefore, we recognize that the absence of sex and age stratification analysis is a limitation of this study.

Additionally, it is important to note that most current GWAS studies have not conducted sex- and age-stratified analyses, which is not unique to our research. In this study, we made efforts to analyze the data from multiple levels (local and global) and angles (GNOVA, HDL, LAVA, cond/conjFDR). Our methodological results were consistent across these approaches, which in part compensates for the lack of sex- and age-stratified analysis.

Furthermore, in the limitations section, we mention that if more sex- and age-stratified data become available, we plan to incorporate sex-specific and age-stratified analyses in future studies to further investigate the role of sex and age in IBD, BMD, and osteoporosis comorbidity (please see lines 412-415 of the revised manuscript).

Lines 412-415: The GWAS datasets used in this study were not stratified by sex and age, limiting our ability to explore sex- and age-specific genetic susceptibility to IBD, BMD, and osteoporosis comorbidity. Future analyses will address this limitation if such data become available.

#2: As the IBD (Crohn’s disease (CD) and ulcerative colitis (UC)) is mentioned in lines 32 and 33, it is expected the immune response genes are involved, but genes that govern the immune response, like HLA genes, or genes related to the absorption and metabolism of calcium, potassium, phosphorus, etc., and genes related to vitamin D, such as VDR, etc., are not analyzed in the study, which may have an important impact on bone density.

It will be a significant addition if further genes are included and their role in bone density is elucidated in a large number of populations and categorized according to their sex and age.

Response:

We sincerely thank the reviewer for the valuable suggestion. We understand that immune response genes (such as HLA genes), genes related to the absorption and metabolism of calcium, potassium, phosphorus, etc., and genes related to vitamin D metabolism (such as VDR) may play an important role in the relationship between IBD and bone density. In our current study, the focus was to establish the overall and local genetic associations between IBD and BMD/osteoporosis through GNOVA, HDL, and LAVA analyses. We also identified shared genetic loci between IBD and BMD/osteoporosis through CondFDR/ConjFDR analysis, including genes like ZBTB40 and RSPO3, which have been repeatedly validated. Therefore, in the discussion section (lines 353-385), we focused on the potential mechanisms of these genes in the context of IBD and BMD/osteoporosis. Based on current literature, ZBTB40 is indeed closely related to immune response, vitamin D metabolism, and calcium ion balance (1,2). RSPO3, on the other hand, is associated with bone density and immune processes (3,4).

Regarding the stratified analysis by sex and age, we have provided a detailed explanation in response to comment #1. We sincerely appreciate the reviewer’s guidance and valuable input.

Reply to Reviewer 2

Abstract

#1:Clarify “biologically significant” loci. Does this mean functional annotation, pathway relevance, or previous association?

Response:

We sincerely thank the reviewer for the valuable question. In this study, “biologically significant” loci refer to those identified based on functional annotation, pathway analysis, and prior associations with traits or diseases. Using conditional/conjunctional false discovery rate (cond/conjFDR) statistical methods, we identified several biologically significant shared genetic loci, which are defined by their functional annotation, relevance to known biological pathways, and prior associations with traits or diseases (5).

Through functional annotation, we found that these loci are enriched in the Wnt signaling pathway. Additionally, we referred to previous studies to confirm that genes such as ZBTB40 and RSPO3 are involved in important biological processes. ZBTB40 plays a critical role in the immune system, vitamin D metabolism, and mineral metabolism, while RSPO3 is closely associated with bone density regulation and immune function. These details are further discussed in the manuscript (lines 353-406).

#2:Add the total number of GWAS datasets or total sample size used for clarity.

Response:

We thank the reviewer for the suggestion. The total number of GWAS datasets and sample size has been added for clarity, as seen in lines 15-16 of the revised manuscript.

Lines 15-16: Utilizing data from genome-wide association studies on 59,957 IBD, 40,266 CD, 45,975 UC, 31,492 BMD, and 399,054 osteoporosis,

Introduction

#1:Update references to include more recent global epidemiological data where possible.

Response:

We sincerely thank the reviewer for the suggestion. In response, we have updated the references with more recent global epidemiological data, as indicated in lines 39-42 and 48-49 of the revised manuscript.

Lines 39-42: Recent 2025 studies show that IBD has entered a "high prevalence stage," with both incidence and prevalence at elevated levels. In Western countries, the prevalence has reached or is near 1%, and by 2030, over 1% of the population is expected to be affected, putting significant strain on healthcare systems (6).

Lines 49-50: Osteoporosis affected 41.5 million people globally in 2019, with a projected 263.2 million cases by 2034 (7).

#2:The paragraph beginning on line 51 is overly dense. Consider splitting into two, and briefly introduce each analytic tool in one sentence before elaborating.

Response:

We sincerely thank the reviewer for the helpful suggestion. In response, we have split the paragraph into three and briefly introduced each analytic tool in one sentence before elaborating. The revisions can be found in lines 54-78 of the revised manuscript.

Methods and Materials

#1:Specify the exact sample size for each GWAS used and any data pre-processing or quality control performed (SNP filtering, ancestry exclusion) (Line 84–91).

Response:

We sincerely thank the reviewer for the helpful comment. In response, we have added the exact sample sizes for each GWAS dataset, including the number of cases, controls, and the total sample size (for BMD, as a continuous variable, we have provided the total sample size instead of case/control counts) in lines 93-100. We have also included Table 1 (lines 695-697), which provides details such as Phenotypic code, Cases/Controls, Ancestry, etc. Furthermore, we have added information on data pre-processing and quality control, including SNP filtering and ancestry exclusion, in lines 101-108.

#2:State the specific genomic resolution used in the LAVA (e.g., 1Mb blocks? predefined regions?). Also, clarify if any Bonferroni or FDR correction was applied in interpreting local correlations (Line 120–130).

Response:

We sincerely thank the reviewer for the insightful comments. In response, we clarify that the specific genomic resolution used in the LAVA analysis is indeed 1Mb blocks (8), as indicated in the revised manuscript (lines 146-147). Additionally, we have included the details regarding the statistical corrections applied to all methods used in this study. These corrections, including Bonferroni and FDR adjustments, are described in lines 109-115 of the manuscript. We hope this addresses your concerns, and we appreciate your helpful feedback.

Lines 109-115: To rigorously control the false-positive rate in the analysis, this study applied specific statistical correction strategies for each method. For the genome-wide genetic correlation analyses conducted using GNOVA, HDL, and MR, we applied the Bonferroni correction. In the case of LAVA analysis, p-values were adjusted using the Benjamini-Hochberg (BH) method to effectively control the false discovery rate (FDR) and identify significant associations. In the condFDR/conjFDR analyses, a significance threshold of conjFDR < 0.05 was adopted, consistent with the practices commonly used in previous studies (9–12).

#3:Clarify whether the FUMA gene annotations were only positional or included eQTL, chromatin interactions, etc.

Response:

We sincerely thank the reviewer for the valuable comment. The FUMA gene annotations were not limited to positional information; they also included gene expression data, chromatin interactions, and other relevant information. In addition, we have provided further details on how the results obtained through conjFDR analysis were annotated using FUMA, which can be found in lines 204-219 of the revised manuscript. We hope this clarification addresses your concerns, and we appreciate your helpful feedback.

Results

#1:Enrichment Results (lines 216–231): Clarify how significance was defined in GO and KEGG enrichment (e.g., FDR < 0.05?).

Response:

We sincerely thank the reviewer for the valuable comment. In response to your suggestion, we have added the methodological details for the GO and KEGG enrichment analysis in the revised manuscript, which can be found in Section 2.7 (lines 220-226). We clarified that significance in the GO and KEGG enrichment analysis was defined using an FDR threshold of < 0.1. We hope this addresses your concern, and we appreciate your helpful feedback.

#2:Were any of the identified shared loci previously known in either disease, or are they novel findings?

Response:

We sincerely thank the reviewer for the valuable question. In our study, some of the identified shared genetic loci have been previously associated with IBD. For example, genes like ZBTB40 and RSPO3 have been shown to be involved in immune response and vitamin D metabolism (1–4). However, we also identified several novel genetic loci, which can be found in Supplementary Table S7-12. In the table, if the “novel gene” column indicates “YES,” it refers to newly identified genes that have not been widely reported in IBD or other related diseases. If the column indicates “NO,” the gene is known. We believe these new findings offer promising directions for further exploration of the genetic mechanisms underlying IBD.

References

#1:Ensure all URLs (e.g., for software tools like FUMA, Sangerbox) are active and appropriately cited. (Line 350-351) “All GWAS data and statistical software utilized in this study were publicly available (accessible via the following URLs)…”

Response:

We sincerely thank the reviewer for the valuable suggestion. We have now included the URLs for accessing the GWAS data and statistical software used in this study (lines 433-442), and have ensured that they are appropriately cited in the manuscript. We hope this addresses your concern, and we appreciate your helpful feedback.

Reply to Reviewer 3

The manuscript by Qin et al. has explored the association between inflammatory bowel disease (IBD) and bone mineral density (BMD) and osteoporosis with extensive statistical tests, and concluded that a notable genetic correlation exists between IBD and BMD/osteoporosis.

The introduction was well written with clear explanation of different statistical methods for GWAS testing and Rg analysis. And the discussion went to great lengths to review the genes/pathways and their potential roles in the development of IBD/BMD/osteoporosis.

However, there are a few questions regarding the design and interpretation of the work, hope the authors could address before considered for publication:

Response:

We sincerely thank the reviewer for their positive feedback and recognition of the clarity in the introduction and thoroughness of the discussion. We also appreciate the constructive points raised regarding the study’s design and interpretation. Below, we provide responses to address the concerns raised and hope the revisions will meet your expectations for publication.

#1:Since the study was solely based on the statistical testing and computational simulation of the GWAS data, could the authors provide more details and characterization of the data? I.e. the sample size, age group, stage of IBD/osteoporosis etc., as well as the features that were selected for statistical testing, which would affect the interpretation of correlation scores.

Response:

We sincerely thank the reviewer for the valuable comment. In our study, the GWAS data used covered 59,957 IBD patients, 40,266 Crohn's disease (CD) patients, 45,97

---

## [Editor Report · Decision Letter 1]

6 Oct 2025

Dear Dr. Qin,

Thank you for submitting your manuscript to PLOS ONE. After careful consideration, we feel that it has merit but does not fully meet PLOS ONE’s publication criteria as it currently stands. Therefore, we invite you to submit a revised version of the manuscript that addresses the points raised during the review process.

We look forward to receiving your revised manuscript.

Kind regards,

Emmanuel O Adewuyi, BPharm, MPH, PhD

Academic Editor

PLOS ONE

Journal Requirements:

Additional Editor Comments:

** ** Thank you for submitting your revised manuscript. Plos One has a standard requirement for reporting MR studies. Please check the details (https://journals.plos.org/plosone/s/best-practices-in-research-reporting#loc-Mendelian-randomization-studies) and strictly follow the guideline. Specifically, authors need to follow STROBE-MR guideline, complete the checklist and include same in their submission. It is also important to include a figure summarising MR method and key assumption. Please, submit all the revision together alongside this new requirement so reviewers can have access to all the responses.

---

## [Author Response · Author response to Decision Letter 2]

8 Oct 2025

RESPONSE LETTER

Oct. 05, 2025

Dear Prof. Emmanuel O Adewuyi, BPharm,

Thank you for reviewing our manuscript (Manuscript ID: PONE-D-25-31485; Manuscript title: A Multilayered Genetic Structure Analysis between Inflammatory Bowel Disease and Bone Density/Osteoporosis) and for your decision of further consideration. Now we have revised and checked the manuscript carefully, according to the editor’s and reviewer’s comments.

We submit here the revised paper. The changes and explanations regarding the revisions of our paper are listed below this letter.

We have tried our best to address all the concerns raised by the editor and reviewer. We hope that, with these modifications and improvements based on the editor’s and reviewer’s comments, our manuscript would be acceptable for publication in your journal.

Thank you again!

With best wishes,

Mengting Qin

Department of Anorectal Surgery, Zhejiang Chinese Medical University Affiliated Jiaxing TCM Hospital, Jiaxing, China

Email: qinmt0907@163.com

Reply to Editor

Thank you for submitting your manuscript to PLOS ONE. After careful consideration, we feel that it has merit but does not fully meet PLOS ONE’s publication criteria as it currently stands. Therefore, we invite you to submit a revised version of the manuscript that addresses the points raised during the review process.

Response:

Thank you for your feedback and for considering our manuscript.

We have carefully reviewed the reviewer's comments and appreciate their valuable feedback. We acknowledge the importance of enhancing the clarity and understanding of our work, and we have already implemented the suggested modifications to address these concerns (please see the point-by-point response below for details).

Thank you again!

ACADEMIC EDITOR:

#1:Reviewers have provided feedback. Authors need to pay close attention and respond to the comments, making corrections to their manuscript as necessary. As noted by some of the reviewers, there is a need to strengthen methodological rigour and provide more detail about the data utilised.

Response:

We sincerely thank the reviewer for the constructive feedback. In response to the comments, we have revised the manuscript to strengthen the methodological clarity and provided additional details regarding the data utilized. The specific changes are summarized below.

#2:Is there a reason the authors did not perform Mendelian randomisation to assess potential causal associations? Yes, I see a reference to previous studies, but could they have assessed this to confirm or contextualise their findings? It would probably add depth and context to their study.

Response:

We sincerely thank the academic editor for this valuable suggestion. We have taken it into account and included a Mendelian randomisation analysis in the revised manuscript (see sections 2.4, lines 161-171, and 3.3, lines 264-276). The results of this analysis are consistent with previous studies, and we believe that this addition has enhanced the depth and context of our findings. We greatly appreciate the editor’s contribution to improving the manuscript.

Reply to Reviewer 1

In the methods:

#1: The total number of participants in the study did not specify the number of males and females or the participants age distribution to identify the more affected sex and age.

Response:

We sincerely thank the reviewer for the valuable suggestion. We have now provided additional details regarding the cohort information in the revised manuscript, including Phenotypes, Phenotypic code, Cases/Controls, and Ancestry (see Table 1, lines 695-697). We acknowledge that the current study did not specify the number of males and females or the participants' age distribution. This is due to the lack of sex- and age-stratified data in our current dataset, which limited our ability to explore sex- and age-specific genetic susceptibility to IBD, BMD, and osteoporosis comorbidity. Therefore, we recognize that the absence of sex and age stratification analysis is a limitation of this study.

Additionally, it is important to note that most current GWAS studies have not conducted sex- and age-stratified analyses, which is not unique to our research. In this study, we made efforts to analyze the data from multiple levels (local and global) and angles (GNOVA, HDL, LAVA, cond/conjFDR). Our methodological results were consistent across these approaches, which in part compensates for the lack of sex- and age-stratified analysis.

Furthermore, in the limitations section, we mention that if more sex- and age-stratified data become available, we plan to incorporate sex-specific and age-stratified analyses in future studies to further investigate the role of sex and age in IBD, BMD, and osteoporosis comorbidity (please see lines 412-415 of the revised manuscript).

Lines 412-415: The GWAS datasets used in this study were not stratified by sex and age, limiting our ability to explore sex- and age-specific genetic susceptibility to IBD, BMD, and osteoporosis comorbidity. Future analyses will address this limitation if such data become available.

#2: As the IBD (Crohn’s disease (CD) and ulcerative colitis (UC)) is mentioned in lines 32 and 33, it is expected the immune response genes are involved, but genes that govern the immune response, like HLA genes, or genes related to the absorption and metabolism of calcium, potassium, phosphorus, etc., and genes related to vitamin D, such as VDR, etc., are not analyzed in the study, which may have an important impact on bone density.

It will be a significant addition if further genes are included and their role in bone density is elucidated in a large number of populations and categorized according to their sex and age.

Response:

We sincerely thank the reviewer for the valuable suggestion. We understand that immune response genes (such as HLA genes), genes related to the absorption and metabolism of calcium, potassium, phosphorus, etc., and genes related to vitamin D metabolism (such as VDR) may play an important role in the relationship between IBD and bone density. In our current study, the focus was to establish the overall and local genetic associations between IBD and BMD/osteoporosis through GNOVA, HDL, and LAVA analyses. We also identified shared genetic loci between IBD and BMD/osteoporosis through CondFDR/ConjFDR analysis, including genes like ZBTB40 and RSPO3, which have been repeatedly validated. Therefore, in the discussion section (lines 353-385), we focused on the potential mechanisms of these genes in the context of IBD and BMD/osteoporosis. Based on current literature, ZBTB40 is indeed closely related to immune response, vitamin D metabolism, and calcium ion balance (1,2). RSPO3, on the other hand, is associated with bone density and immune processes (3,4).

Regarding the stratified analysis by sex and age, we have provided a detailed explanation in response to comment #1. We sincerely appreciate the reviewer’s guidance and valuable input.

Reply to Reviewer 2

Abstract

#1:Clarify “biologically significant” loci. Does this mean functional annotation, pathway relevance, or previous association?

Response:

We sincerely thank the reviewer for the valuable question. In this study, “biologically significant” loci refer to those identified based on functional annotation, pathway analysis, and prior associations with traits or diseases. Using conditional/conjunctional false discovery rate (cond/conjFDR) statistical methods, we identified several biologically significant shared genetic loci, which are defined by their functional annotation, relevance to known biological pathways, and prior associations with traits or diseases (5).

Through functional annotation, we found that these loci are enriched in the Wnt signaling pathway. Additionally, we referred to previous studies to confirm that genes such as ZBTB40 and RSPO3 are involved in important biological processes. ZBTB40 plays a critical role in the immune system, vitamin D metabolism, and mineral metabolism, while RSPO3 is closely associated with bone density regulation and immune function. These details are further discussed in the manuscript (lines 353-406).

#2:Add the total number of GWAS datasets or total sample size used for clarity.

Response:

We thank the reviewer for the suggestion. The total number of GWAS datasets and sample size has been added for clarity, as seen in lines 15-16 of the revised manuscript.

Lines 15-16: Utilizing data from genome-wide association studies on 59,957 IBD, 40,266 CD, 45,975 UC, 31,492 BMD, and 399,054 osteoporosis,

Introduction

#1:Update references to include more recent global epidemiological data where possible.

Response:

We sincerely thank the reviewer for the suggestion. In response, we have updated the references with more recent global epidemiological data, as indicated in lines 39-42 and 48-49 of the revised manuscript.

Lines 39-42: Recent 2025 studies show that IBD has entered a "high prevalence stage," with both incidence and prevalence at elevated levels. In Western countries, the prevalence has reached or is near 1%, and by 2030, over 1% of the population is expected to be affected, putting significant strain on healthcare systems (6).

Lines 49-50: Osteoporosis affected 41.5 million people globally in 2019, with a projected 263.2 million cases by 2034 (7).

#2:The paragraph beginning on line 51 is overly dense. Consider splitting into two, and briefly introduce each analytic tool in one sentence before elaborating.

Response:

We sincerely thank the reviewer for the helpful suggestion. In response, we have split the paragraph into three and briefly introduced each analytic tool in one sentence before elaborating. The revisions can be found in lines 54-78 of the revised manuscript.

Methods and Materials

#1:Specify the exact sample size for each GWAS used and any data pre-processing or quality control performed (SNP filtering, ancestry exclusion) (Line 84–91).

Response:

We sincerely thank the reviewer for the helpful comment. In response, we have added the exact sample sizes for each GWAS dataset, including the number of cases, controls, and the total sample size (for BMD, as a continuous variable, we have provided the total sample size instead of case/control counts) in lines 93-100. We have also included Table 1 (lines 695-697), which provides details such as Phenotypic code, Cases/Controls, Ancestry, etc. Furthermore, we have added information on data pre-processing and quality control, including SNP filtering and ancestry exclusion, in lines 101-108.

#2:State the specific genomic resolution used in the LAVA (e.g., 1Mb blocks? predefined regions?). Also, clarify if any Bonferroni or FDR correction was applied in interpreting local correlations (Line 120–130).

Response:

We sincerely thank the reviewer for the insightful comments. In response, we clarify that the specific genomic resolution used in the LAVA analysis is indeed 1Mb blocks (8), as indicated in the revised manuscript (lines 146-147). Additionally, we have included the details regarding the statistical corrections applied to all methods used in this study. These corrections, including Bonferroni and FDR adjustments, are described in lines 109-115 of the manuscript. We hope this addresses your concerns, and we appreciate your helpful feedback.

Lines 109-115: To rigorously control the false-positive rate in the analysis, this study applied specific statistical correction strategies for each method. For the genome-wide genetic correlation analyses conducted using GNOVA, HDL, and MR, we applied the Bonferroni correction. In the case of LAVA analysis, p-values were adjusted using the Benjamini-Hochberg (BH) method to effectively control the false discovery rate (FDR) and identify significant associations. In the condFDR/conjFDR analyses, a significance threshold of conjFDR < 0.05 was adopted, consistent with the practices commonly used in previous studies (9–12).

#3:Clarify whether the FUMA gene annotations were only positional or included eQTL, chromatin interactions, etc.

Response:

We sincerely thank the reviewer for the valuable comment. The FUMA gene annotations were not limited to positional information; they also included gene expression data, chromatin interactions, and other relevant information. In addition, we have provided further details on how the results obtained through conjFDR analysis were annotated using FUMA, which can be found in lines 204-219 of the revised manuscript. We hope this clarification addresses your concerns, and we appreciate your helpful feedback.

Results

#1:Enrichment Results (lines 216–231): Clarify how significance was defined in GO and KEGG enrichment (e.g., FDR < 0.05?).

Response:

We sincerely thank the reviewer for the valuable comment. In response to your suggestion, we have added the methodological details for the GO and KEGG enrichment analysis in the revised manuscript, which can be found in Section 2.7 (lines 220-226). We clarified that significance in the GO and KEGG enrichment analysis was defined using an FDR threshold of < 0.1. We hope this addresses your concern, and we appreciate your helpful feedback.

#2:Were any of the identified shared loci previously known in either disease, or are they novel findings?

Response:

We sincerely thank the reviewer for the valuable question. In our study, some of the identified shared genetic loci have been previously associated with IBD. For example, genes like ZBTB40 and RSPO3 have been shown to be involved in immune response and vitamin D metabolism (1–4). However, we also identified several novel genetic loci, which can be found in Supplementary Table S7-12. In the table, if the “novel gene” column indicates “YES,” it refers to newly identified genes that have not been widely reported in IBD or other related diseases. If the column indicates “NO,” the gene is known. We believe these new findings offer promising directions for further exploration of the genetic mechanisms underlying IBD.

References

#1:Ensure all URLs (e.g., for software tools like FUMA, Sangerbox) are active and appropriately cited. (Line 350-351) “All GWAS data and statistical software utilized in this study were publicly available (accessible via the following URLs)…”

Response:

We sincerely thank the reviewer for the valuable suggestion. We have now included the URLs for accessing the GWAS data and statistical software used in this study (lines 433-442), and have ensured that they are appropriately cited in the manuscript. We hope this addresses your concern, and we appreciate your helpful feedback.

Reply to Reviewer 3

The manuscript by Qin et al. has explored the association between inflammatory bowel disease (IBD) and bone mineral density (BMD) and osteoporosis with extensive statistical tests, and concluded that a notable genetic correlation exists between IBD and BMD/osteoporosis.

The introduction was well written with clear explanation of different statistical methods for GWAS testing and Rg analysis. And the discussion went to great lengths to review the genes/pathways and their potential roles in the development of IBD/BMD/osteoporosis.

However, there are a few questions regarding the design and interpretation of the work, hope the authors could address before considered for publication:

Response:

We sincerely thank the reviewer for their positive feedback and recognition of the clarity in the introduction and thoroughness of the discussion. We also appreciate the constructive points raised regarding the study’s design and interpretation. Below, we provide responses to address the concerns raised and hope the revisions will meet your expectations for publication.

#1:Since the study was solely based on the statistical testing and computational simulation of the GWAS data, could the authors provide more details and characterization of the data? I.e. the sample size, age group, stage of IBD/osteoporosis etc., as well as the features that were selected for statistical testing, which would affect the interpretation of correlation scores.

Response:

We sincerely thank the reviewer for the valuable comment. In our study, the GWAS data used covered 59,957 IBD patients, 40,266 Crohn's disease (CD) patients, 45,97

---

## [Decision Letter · Decision Letter 2]

30 Oct 2025

A Multilayered Genetic Structure Analysis between Inflammatory Bowel Disease and Bone Density/Osteoporosis

PONE-D-25-31485R2

Dear Dr. Qin,

We’re pleased to inform you that your manuscript has been judged scientifically suitable for publication and will be formally accepted for publication once it meets all outstanding technical requirements.

Kind regards,

Emmanuel O Adewuyi, BPharm, MPH, PhD

Academic Editor

PLOS ONE

Additional Editor Comments (optional):

Reviewers' comments:

Reviewer's Responses to Questions

**Comments to the Author**

Reviewer #1: All comments have been addressed

Reviewer #2: All comments have been addressed

Reviewer #3: All comments have been addressed

Reviewer #4: All comments have been addressed

Reviewer #5: All comments have been addressed

2. Is the manuscript technically sound, and do the data support the conclusions?

Reviewer #1: Yes

Reviewer #2: Yes

Reviewer #3: Yes

Reviewer #4: Yes

Reviewer #5: (No Response)

3. Has the statistical analysis been performed appropriately and rigorously?

Reviewer #1: N/A

Reviewer #2: Yes

Reviewer #3: Yes

Reviewer #4: Yes

Reviewer #5: (No Response)

4. Have the authors made all data underlying the findings in their manuscript fully available?

Reviewer #1: Yes

Reviewer #2: Yes

Reviewer #3: Yes

Reviewer #4: Yes

Reviewer #5: (No Response)

5. Is the manuscript presented in an intelligible fashion and written in standard English?

Reviewer #1: Yes

Reviewer #2: Yes

Reviewer #3: Yes

Reviewer #4: Yes

Reviewer #5: (No Response)

Reviewer #1: To the authors,

The study represents an analysis of huge genetics data to predict the association between inflammatory bowel disease (IBD) and reduced bone mineral density (BMD) or osteoporosis. It opens new insight into new understanding of the associated genetic status in patients with inflammatory disease, which may lead to abnormal bone density or osteoporosis. Thanks for your efforts.

Reviewer #2: Great work by the authors; all my comments have been addressed, and I recommend the study to be accepted.

Reviewer #3: The authors have addressed the comments well with good reasoning and additional tests. No further comment.

Reviewer #4: The authors have successfully addressed all my previous concerns and I don't have any further questions.

Reviewer #5: (No Response)

**Do you want your identity to be public for this peer review?** For information about this choice, including consent withdrawal, please see our Privacy Policy

Reviewer #1: **Yes: ** Luma Hassan Alwan Al Obaidy

Reviewer #2: No

Reviewer #3: No

Reviewer #4: No

Reviewer #5: No

---

## [Editor Report · Acceptance letter]

PONE-D-25-31485R2

PLOS ONE

Dear Dr. Qin,

I'm pleased to inform you that your manuscript has been deemed suitable for publication in PLOS ONE. Congratulations! Your manuscript is now being handed over to our production team.

Kind regards,

on behalf of

Dr. Emmanuel O Adewuyi

Academic Editor

PLOS ONE